# Multicenter evaluation of label-free quantification in human plasma on a high dynamic range benchmark set

Human plasma is routinely collected during clinical care and constitutes a rich source of biomarkers for diagnostics and patient stratification. Liquid chromatography-mass spectrometry (LC-MS)-based proteomics is a key method for plasma biomarker discovery, but the high dynamic range of plasma proteins poses significant challenges for MS analysis and data processing. To benchmark the quantitative performance of neat plasma analysis, we introduce a multispecies sample set based on a human tryptic plasma digest containing varying low level spike-ins of yeast and *E. coli* tryptic proteome digests, termed PYE. By analysing the sample set on state-of-the-art LC-MS platforms across twelve different sites in data-dependent (DDA) and data-independent acquisition (DIA) modes, we provide a data resource comprising a total of 1116 individual LC-MS runs. Centralized data analysis shows that DIA methods outperform DDA-based approaches regarding identifications, data completeness, accuracy, and precision. DIA achieves excellent technical reproducibility, as demonstrated by coefficients of variation (CVs) between 3.3% and 9.8% at protein level. Comparative analysis of different setups clearly shows a high overlap in identified proteins and proves that accurate and precise quantitative measurements are feasible across multiple sites, even in a complex matrix such as plasma, using state-of-the-art instrumentation. The collected dataset, including the PYE sample set and strategy presented, serves as a valuable resource for optimizing the accuracy and reproducibility of LC-MS and bioinformatic workflows for clinical plasma proteome analysis.

Human blood and blood-derived components (i.e., serum and plasma) reflect an individual´s health state and are routinely used for in vitro diagnostics, often referred to as a liquid biopsy, to either monitor, detect, predict, or rule out diseases. Plasma, the liquid blood component, is obtained by removing cellular material from whole blood through centrifugation in the presence of anti-coagulants such as heparin, ethylenediaminetetraacetic acid (EDTA), or sodium citrate. Plasma and serum are the most collected biofluids globally, easily accessible and routinely taken from thousands of patients daily. As such, they are valuable sources of (bio)markers reflecting the states of various disorders and illnesses and has become the focus of pharmacological, biomedical, and clinical pursuits.

The vast majority of biological processes are controlled and carried out by proteins. Liquid chromatography-mass spectrometry (LC-MS) has evolved as the leading technology for investigating proteins and analysing entire proteomes across diverse biological systems, making it a powerful tool for (protein) biomarker discovery[1,2]. Within their detection limits, MS-based proteomic approaches allow for the unbiased and comprehensive characterization of all proteins in a system with high analytical specificity. Most of these workflows employ a

✉e-mail: ute.distler@uni-mainz.de; tenzer@uni-mainz.de

bottom-up approach, where sample proteins are first digested in vitro with sequence-specific proteases, such as trypsin, to generate peptides for analysis. Despite tremendous technological advances in the field of MS over the past two decades, plasma proteome analysis by this technology remains challenging due to the extremely high dynamic range of plasma proteins, which spans over 11 orders of magnitude[3,4]. Albumin, the most abundant plasma protein at a concentration of ~70 mg/mL, constitutes around 55% of the total plasma protein content, while the 22 most abundant proteins collectively account for 99% of the overall plasma protein mass[3,4]. In MS-based bottom-up proteomic workflows, the majority of quantified peptide intensities arises from these highly abundant plasma proteins, significantly hindering the detection and quantification of peptides derived from lower-abundance proteins. As a result, in typical MS analyses of neat plasma, only a few hundred classical plasma proteins can be reliably detected and quantified across multiple studies[4,5]. These include proteins with a functional role in blood such as albumin, apolipoproteins, immunoglobulins, and acute phase proteins, as well as members of the coagulation cascade. Lower-abundance proteins, including those derived from tissue leakage or signaling proteins such as cytokines, often fall outside the dynamic range of detection spanning ~4–5 orders of magnitude on most of the current generation instrument platforms[4]. Even when detected, quantifying low-abundant plasma proteins remains challenging, as they are prone to lower signal-to-noise ratios, poor ion statistics, and missing (peptide intensity) values across runs, all of which contribute to higher variance and reduced quantitation precision and accuracy[6–8].

Over the past two decades, significant efforts have been made to reduce the dynamic range of plasma samples and enhance the depth of plasma proteome coverage. Strategies such as immunoaffinity-based depletion of abundant proteins[9–11], selective precipitation[12], nanoparticle-assisted enrichment[13–15] and magnetic bead-based isolation of plasma extracellular vesicles[16] enabled the identification of up to ~4500 proteins in plasma. Despite their advantages, these methods are often constrained by high costs, limited throughput, and technique-specific biases[17,18]. Consequently, analysis of neat plasma continues to be a commonly used approach in proteomic studies.

In clinical contexts, achieving accurate and reproducible quantification is essential. The discovery and verification of potential biomarkers depend heavily on the dynamic range, accuracy, and precision of quantitative measurements across large cohorts, multiple platforms, and study centers. Over the past years, several intra- and interlaboratory studies have addressed this issue using distinct benchmark sample sets to assess quantitative reproducibility of different (label-free) proteomic LC-MS workflows or data analysis tools[7,8,19–23]. Such benchmark samples can be generated by spiking synthetic peptides or proteins into a matrix at known amounts[20–23], mixing whole proteomes at distinct ratios[7,8,19,24–26] or a combination of both[27]. Common to these sample sets is that they represent a ground truth and allow either to optimize different steps of an LC-MS workflow, assess its qualitative and quantitative performance[24], or conduct cross-center comparisons[20,28]. Hence, these samples are widely used, e.g., for comparing software tools and data analysis workflows, as they facilitate the selection of the best-performing quantitative data analysis pipeline for distinct LC-MS setups[6,8,29]. Moreover, they allow the evaluation of novel MS hardware[30], facilitate the benchmarking of software for data analysis[31,32], and help optimize (data) processing algorithms to improve quantitative precision and accuracy[7]. Additionally, they are a valuable tool for multilaboratory[20] and cross-platform comparisons[26,29,30], providing a snapshot of the technological landscape and workflow performance at the respective study timepoint. Recently, Fröhlich et al.[25] introduced a mixed proteome dataset designed to incorporate real-world inter-patient heterogeneity, enabling the benchmarking of data-independent acquisition (DIA) data analysis workflows in clinical settings, particularly for formalin-fixed

paraffin-embedded tissue samples. However, a ground truth benchmark set specifically for assessing quantitative accuracy and precision in neat plasma analysis has yet to be established. Recently, the CLINSPECT-M consortium, part of the German MSCoreSys clinical proteomics initiative, initiated a round-robin study among its six proteomic laboratories assessing current best practices for sample preparation and LC-MS measurement for clinically relevant body fluids such as plasma and cerebrospinal fluid[33].

In this work, we complement this effort by evaluating the quantitative performance of neat plasma analysis across twelve different partner sites of the MSCoreSys clinical proteomics research consortium (https://www.mscoresys.de/), including different state-of-the-art LC-MS instrument platforms. To this end, we introduce a benchmark set of six samples based on a human tryptic plasma digest containing varying amounts of tryptic digests of yeast and *Escherichia coli* proteomes (PYE). The PYE benchmark set is an evolution of the hybrid proteome sample set initially described by Kuharev et al.[19] and Navarro et al.[7], addressing the challenges posed by the high dynamic protein range typical for neat plasma. Each participating site received and measured the PYE sample set on their respective LC-MS platforms using data-dependent acquisition (DDA)- and/or DIA-based methods. Importantly, no particular guidelines, protocols, or restrictions were enforced. All generated raw data have been centrally analysed through a unified pipeline, using MaxQuant[34,35] for DDA and DIA-NN[36] for DIA data. The resulting dataset clearly demonstrates that accurate and precise protein quantification applying state-of-the-art MS-based proteomics is achievable, even within the complex plasma matrix, across different instrument platforms and multiple sites when applying DIA-based approaches.

## Results
### Study design and PYE benchmark sample set
The aim of the present study was to assess and benchmark qualitative and quantitative reproducibility as well as the accuracy and precision across multiple sites and instrument platforms using a benchmark sample set that addresses the challenges of protein dynamic range in neat plasma. To this end, we defined a multispecies sample set based on a human tryptic plasma digest, containing varying spike-in levels of tryptic-digested yeast and *E. coli* (PYE) proteomes. The PYE benchmark set comprises six samples in total: PYE1 A and B, PYE3 A and B, PYE9 A and B. In these samples, human plasma digest serves as a high dynamic range background, whereas low-level spike-ins of *E. coli* and yeast tryptic peptides mimic regulated proteins between two samples, A and B, allowing to evaluate precision and accuracy of label-free quantification. In samples PYE1 A and B, human plasma proteins account for 90% of the total protein mass, and yeast and *E. coli* proteins for the remaining 10% (Fig. 1a). Tryptic peptides were combined in the following ratios: sample PYE A contains 90% w/w human, 2% w/w yeast, and 8% w/w *E. coli* proteins. Sample PYE B is composed of 90% w/w human, 6% w/w yeast, and 4% w/w *E. coli* proteins. To simulate the challenges of protein dynamic range in clinical plasma samples, the samples PYE1 A and B were further diluted using tryptically digested human plasma, thus additionally reducing the spike-in levels of yeast and *E.coli* digests (see Fig. 1a). PYE3 refers to a 1:3 and PYE9 to a 1:9 dilution of the PYE1 sample set, with PYE9 containing only 1.1% of non-human proteins. The samples were centrally prepared and shipped to all participating sites on dry ice. Shipped sample amounts depended on the LC-MS setup used at the respective site. Per setup, all samples were to be analysed in six replicate injections. Additionally, two blank injections had to be performed prior to the sample runs to avoid carry-over from system quality control runs, typically conducted using HeLa or K562 tryptic digests (see also method section). MS raw data files were uploaded and analysed centrally using either MaxQuant, for DDA, or DIA-NN, for DIA data.

In total, twelve study centers of the MSCoreSys consortium (sites A to L; for an overview on site specific setups see Table 1) took part in the round robin study, collecting 34 full PYE data sets (most of them, with a

few exceptions, comprising six replicate measurements of samples PYE1 A, PYE1 B, PYE3 A, PYE3 B, PYE9 A, and PYE9 B, see Table 1 and Supplementary Data 1). Measurements were conducted on different instrument platforms in either DDA and/or DIA mode, encompassing 1116 individual LC-MS runs. Overall, 13 DDA and DIA data sets were acquired using the exact same LC-MS setup, allowing a direct comparison of both acquisition modes. Mass spectrometers from various manufacturers were used in the present study for data collection, including instruments from ThermoFisher (Orbitrap Eclipse, Orbitrap Exploris 480, Orbitrap Fusion Lumos, Q Exactive HF, Q Exactive HF-X), Bruker (timsTOF Pro, timsTOF Pro2) and Sciex (zenoTOF). In total, seven different LC platforms were used for peptide separation prior to MS analysis, including the following models, Ultimate 3000, Vanquish Neo and EASY-nLC 1200 from ThermoFisher, Evosep One (Evosep), nanoElute (Bruker), nanoAcquity and M-Class from Waters Corporation. Most of the LC systems were operated in the nanoflow range, four sites (sites D, E, F, and K, see Table 1) included micro-flow LC-MS/MS analyses on their Vanquish Neo LC and M-Class systems. Overall, 13 different LC-MS setups were used, with the Ultimate 3000 being the predominant LC platform and the Orbitrap Exploris 480 the prevalent MS instrument in this study (see Fig. 1b, Supplementary Data 1).

## PYE proteome coverage depends on PYE dilution, MS acquisition mode, overall analysis time, LC-MS setup and data processing software

To compare the performance of the different LC-MS setups, we first evaluated the number of proteins and peptides that were identified in each setting and sample (see Fig. 2a, b, Supplementary Figs. 1 and 2, Supplementary Data 2). Overall, we observed a high variability in protein and peptide identifications (IDs) between the different LC-MS setups and acquisition modes as exemplarily shown for PYE1 (Fig. 2a, Supplementary Fig. 1a, Supplementary Data 2). IDs were markedly lower for the DDA as compared to the DIA datasets: In case of DDA, IDs ranged from 919 to 2759 protein groups (1743 protein groups and 15,835 peptides on average), whereas numbers of identified protein groups varied between 1433 and 4653 (with an average of 3193 detected proteins and 29,259 peptides) in case of DIA. Moreover, DIA approaches demonstrated superior reproducibility in terms of identified proteins and peptides, as exemplarily illustrated for the PYE1 A/B set. On average, 84.2% of proteins were consistently identified across all runs within each DIA setup, while this was the case for only 51.5% of proteins (on average) within a DDA setup (Fig. 2a, Supplementary Data 2).

Besides the acquisition mode, the number of identified proteins also depended on the analysis time, i.e., gradient length. For example, the DIA dataset with the lowest number of IDs (L_nAcqu_tTOF) was acquired running an 11 min gradient, whereas the gradient length was 102 min for the setup with the highest protein IDs (H_ulti_ex). Many sites, however, used similar gradient lengths for the LC-MS analyses ranging either between 29 and 48 min or around 60 and 70 min for DIA and mainly around and above 50 min for DDA analyses. Interestingly, averaging the ID numbers, we did not observe marked differences between setups with a gradient length of 29–48 min (3235 protein groups) and 60–70 min (3039 protein groups) in DIA mode. However,

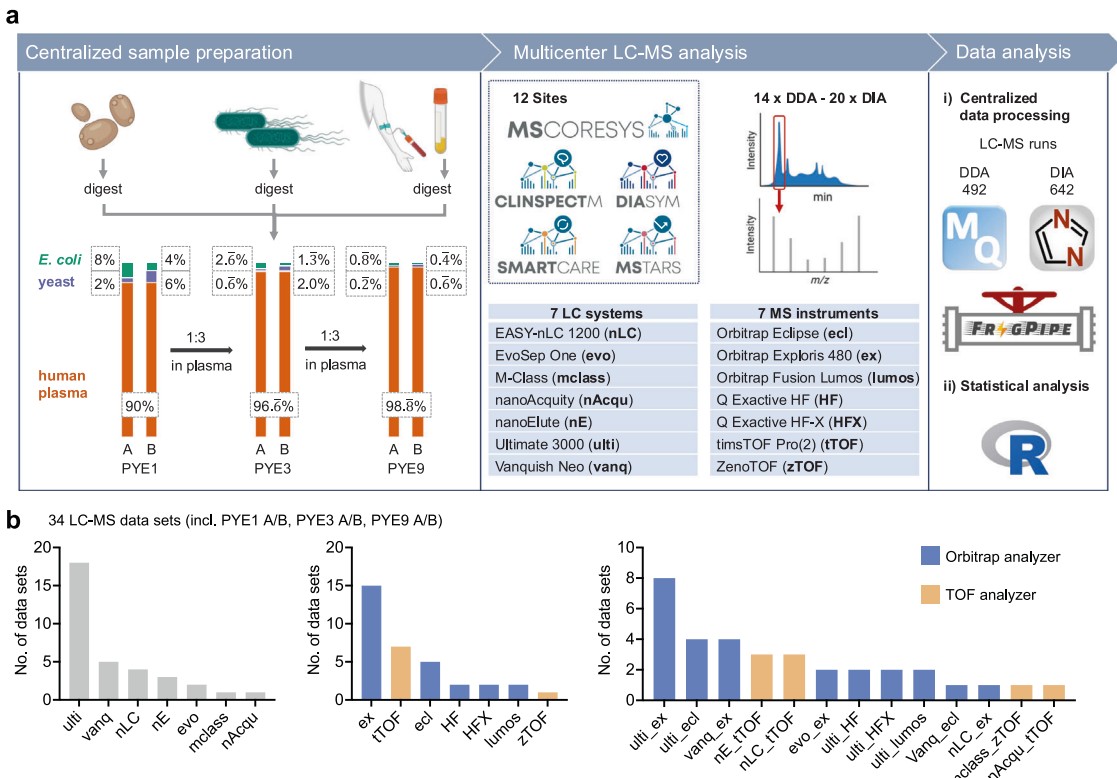

**Fig. 1 | Overview of the PYE sample set and the study design. a** Left panel: The PYE sample set was centrally prepared and consists of six different samples. A tryptic digest of human plasma (orange) serves as background. Varying spike-ins of tryptic *E. coli* (green) and yeast proteomes (violet) mimic differentially regulated proteins between samples with the denominations A and B enabling the evaluation of label-free quantification in a plasma background. Exact sample compositions are provided in the black boxes (percent of total protein mass). To resemble the challenges of protein dynamic range in clinical plasma samples, sample set PYE1 was further diluted with tryptic human plasma reducing the proportion of tryptic *E. coli* and yeast proteomes in the sample sets PYE3 and PYE9. Middle panel: PYE samples were shipped for LC-MS analysis to twelve different sites of the MSCoreSys network. Right panel: Subsequent raw data and statistical analyses of all acquired data sets were conducted centrally. **b** Overview of instrumentation and LC-MS setups used in the round robin study. Parts of **a** partially generated in Biorender (https://BioRender.com/9vhkffx, https://creativecommons.org/licenses/by-sa/4.0/ for R logo).

**Table 1 | Overview of the collected datasets in the present multicenter study**

| Lab ID | LC system | MS | Gradient length [min] | DIA[a] | DDA[a] |
|--------|-----------|-----|------------------|--------|--------|
| A | Ultimate 3000 | Exploris | 50 | yes (6) | yes (6) |
| A | EvoSep | Exploris | 44 | yes (6) | yes (6) |
| B | Ultimate 3000 | HF | 90 | yes (6) | yes (6) |
| B | Ultimate 3000 | HFX | 90 | yes (6) | yes (6) |
| C | nanoElute | timsTOF Pro | 70 | yes (6)[b] | yes (6) |
| D | Ultimate 3000 | Orbitrap Eclipse | 60 | yes (6) | yes (6) |
| D | Vanquish Neo (MF) | Exploris | 60 | yes (6) | yes (6) |
| E | Ultimate 3000 | Fusion Lumos | 60 | yes (6) | yes (6) |
| E | Vanquish Neo (MF) | Exploris | 60 | yes (5) | yes (5) |
| F | Vanquish Neo (MF) | Orbitrap Eclipse | 60 | | yes (6) |
| G | Easy nLC 1200 | timsTOF | 30 | yes (6) | yes (6) |
| G | Ultimate 3000 | Exploris | 48 | yes (6) | yes (6) |
| H | Ultimate 3000 | Orbitrap Eclipse | 120 | yes (6) | yes (6) |
| H | Ultimate 3000 | Exploris | 102 | yes (6) | yes (6) |
| I | Easy nLC 1200 | Exploris | 44 | yes (6) | |
| J | Easy nLC 1200 | timsTOF | 44 | yes (3) | |
| K | M-Class (MF) | ZenoTOF | 20 | yes (6) | |
| L | nanoElute | timsTOF Pro2 | 35.5 | yes (3) | |
| L | nanoAcquitiy | timsTOF Pro | 11 | yes (3) | |
| L | Ultimate 3000 | Exploris FAIMS | 29 | yes (3) | |
| L | Ultimate 3000 | Exploris | 29 | yes (3) | |

[a]Numbers in brackets indicate the numbers of replicate measurements conducted, i.e., three to six replicates of each PYE sample were acquired per laboratory setting.
[b]One raw data file (C_nE_tTOF PYE3 B replicate 1 DIA) had to be excluded, as number of identifications in DIA-NN were below 60% of average as compared to the remaining replicates of this sample and setup.

for some DIA setups with similar analysis times, we observed marked differences in the protein ID rate, i.e., proteins identified in relation to gradient length (see Fig. 2a). This can likely be attributed to the lab-specific differences in instrumentation and LC-MS method settings. For example, most of the TOF datasets were acquired using 29–48 min gradients, while the 60–70 min datasets constitute mainly Orbitrap data. Among the 60–70 min datasets the two microflow setups (D_Vanq_ex and E_Vanq_ex) show slightly lower protein IDs (on average around 2400 proteins) as compared to the other setups with similar gradient length (averaging 3465 protein groups). In contrast to our expectations, we observed no significant systematic influence of peak capacity, cycle time, or signal response on the number of identifications. Overall, we found an overlap of 683 proteins (from a total of 3506 proteins) that were identified in all DDA datasets and 928 out of 5785 proteins that were shared across all DIA runs for PYE1. Over 1600 proteins were shared in 90% of DIA datasets, i.e., across 18 setups. Moreover, 541 proteins were consistently detected in all 34 LC-MS setups (Fig. 2c, d, Supplementary Fig. 3). These numbers are, of course, impacted by setups with lower proteome coverage. When comparing different instrument setups with similar coverage or those with fewer IDs to those with a deeper proteome coverage, we observed a significant overlap of identified proteins, reaching in many cases up to 80–90% (Supplementary Fig. 3), highlighting the reproducibility of LC-MS based plasma proteomic analyses across different labs.

The choice of processing software can significantly impact the number of peptide and protein IDs, owing to differences in search and protein inference algorithms. To assess the influence of software on IDs and to process both, the DIA and DDA data, with the same tool, we additionally analysed the whole round robin dataset with the latest version of FragPipe (version 23, see Supplementary Figs. 4–6). In case of the DDA analyses, a marked increase in proteome coverage and reproducibility was observed, as reflected by an enhanced overlap among technical replicates and across distinct LC-MS instrumentation setups compared to the MaxQuant results. In contrast, proteome coverage was markedly lower for DIA as compared to the DIA-NN analysis, which on average yielded around 25% more protein IDs compared to FragPipe. Hence, the gap between DDA and DIA is by far not as prevalent when processing the dataset in FragPipe with some matching setups showing similar numbers of IDs. Nevertheless, on average, IDs were higher in DIA mode (around 17%) comparing all matching DDA and DIA runs. Of note, IDs across the different LC-MS setups show similar patterns as compared to MaxQuant and DIA-NN, with the same setups achieving highest and lowest numbers of IDs, respectively.

Across all settings, the highest number of proteins was consistently identified in PYE1 A/B as compared to PYE3 A/B and PYE9 A/B samples, which is to be expected as the percentage of *E. coli* and yeast proteins is highest in the PYE1 set. Regarding species-specific IDs, the numbers of detected human plasma proteins were similar between PYE1, PYE3, and PYE9 within each setting, while we observed a marked drop in IDs for *E. coli* and yeast proteins from PYE1 to PYE3 and PYE9 (Fig. 2b, Supplementary Data 3). Independent of the LC-MS setting used, a three-fold reduction of spike-in levels of *E. coli* and yeast tryptic digests reduced the number of *E. coli* and yeast protein IDs around 1.85-fold in DDA and 1.7-fold in DIA mode between PYE1 and PYE3 and around 2.35- (DDA) as well as 2-fold (DIA) between PYE3 and PYE9, respectively.

This is also reflected when integrating the results from all DDA and DIA datasets across the different sites (Fig. 3a, b). For both, DDA and DIA mode, the dynamic range of identified proteins is similar between PYE1, PYE3, and PYE9, spanning four orders of magnitude in the case of each species, except for human plasma proteins identified by DIA which cover six orders of magnitude. However, with each dilution step from PYE1 to PYE9, a distinct number of *E. coli* and yeast proteins falls below the detection limit, resulting in a reduced proteome coverage for both, DDA and DIA datasets. In DIA mode, we observed a 1.3- (*E. coli*) to 1.4-fold (yeast) decrease in protein IDs in PYE3 and a 2.0- (*E. coli*) to 2.5-fold (yeast) decrease in PYE9 as compared PYE1. In case of DDA, the drop was slightly higher. Here, ID numbers decreased by factors of around 1.6 in case of PYE3 and 2.6 for PYE9 as compared to PYE1 for both yeast and *E. coli* proteins. Overall, abundances of commonly identified proteins show a high correlation for both acquisition modes between the PYE1, PYE3 and PYE9 sample sets (Fig. 3c, d). As anticipated from the serial dilution between sample sets, point clouds pertaining to *E. coli* and yeast proteins center around the expected ratios indicated by the dotted lines.

Notably, the design of the PYE sample additionally allows to determine the lower limit of detection (LOD) and linearity for thousands of analytes as a function of their signal intensities by comparing label-free quantification (LFQ) values of individual proteins of *E.coli* spike-ins across six dilution levels, covering a 18-fold difference between PYE_1A and PYE_9B (Fig. 3e, f). Overall, both DDA and DIA showed good linearity across all six samples. In addition, our analysis revealed that the 10% lowest abundant *E.coli* proteins (as defined by a low LFQ value in PYE1) already fall below detection limit in the PYE3_A sample in DDA, while they remain detectable in both PYE3_A and PYE3_B samples in DIA mode, indicating a lower LOD for DIA quantification.

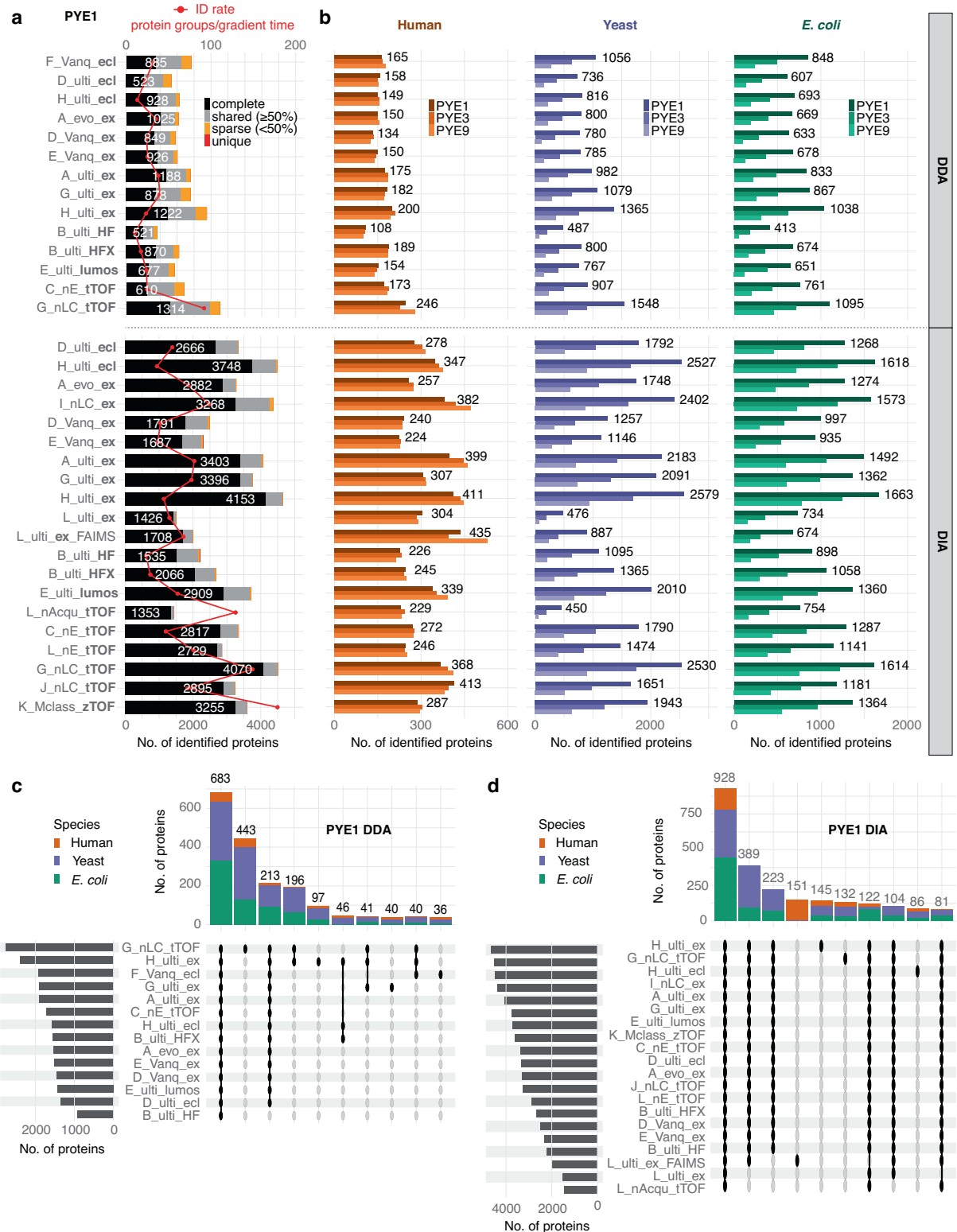

**Fig. 2 | Number of identified proteins in the PYE sample set for different LC-MS setups and sites. a** Number of identified protein groups in the PYE1 sample set for the different setups. Colours indicate the number of proteins identified in all replicate runs per setup (complete, black), equal and more than 50% of runs (grey) as well as sparse (below 50%, orange) and unique identifications (red). White numbers indicate proteins identified in all replicate runs and red dots the number of identified proteins in relation to the programmed gradient length, i.e., protein IDs per min. Letters refer to the sites (Supplementary Data 2). **b** Number of

identified protein groups across the whole PYE dataset (PYE1, PYE3, PYE9) for each setup split by species (human: orange, yeast: violet, *E. coli*: green). Numbers refer to proteins identified in PYE1 (Supplementary Data 3). Upset plots showing the overlap of identified proteins in the PYE1 sample set by **c** DDA- and **d** DIA-based approaches across multiple sites and LC-MS platforms. Proportions of human (orange), yeast (violet) and *E. coli* (green) proteins are indicated within the bars. Source data are provided as a Source Data file.

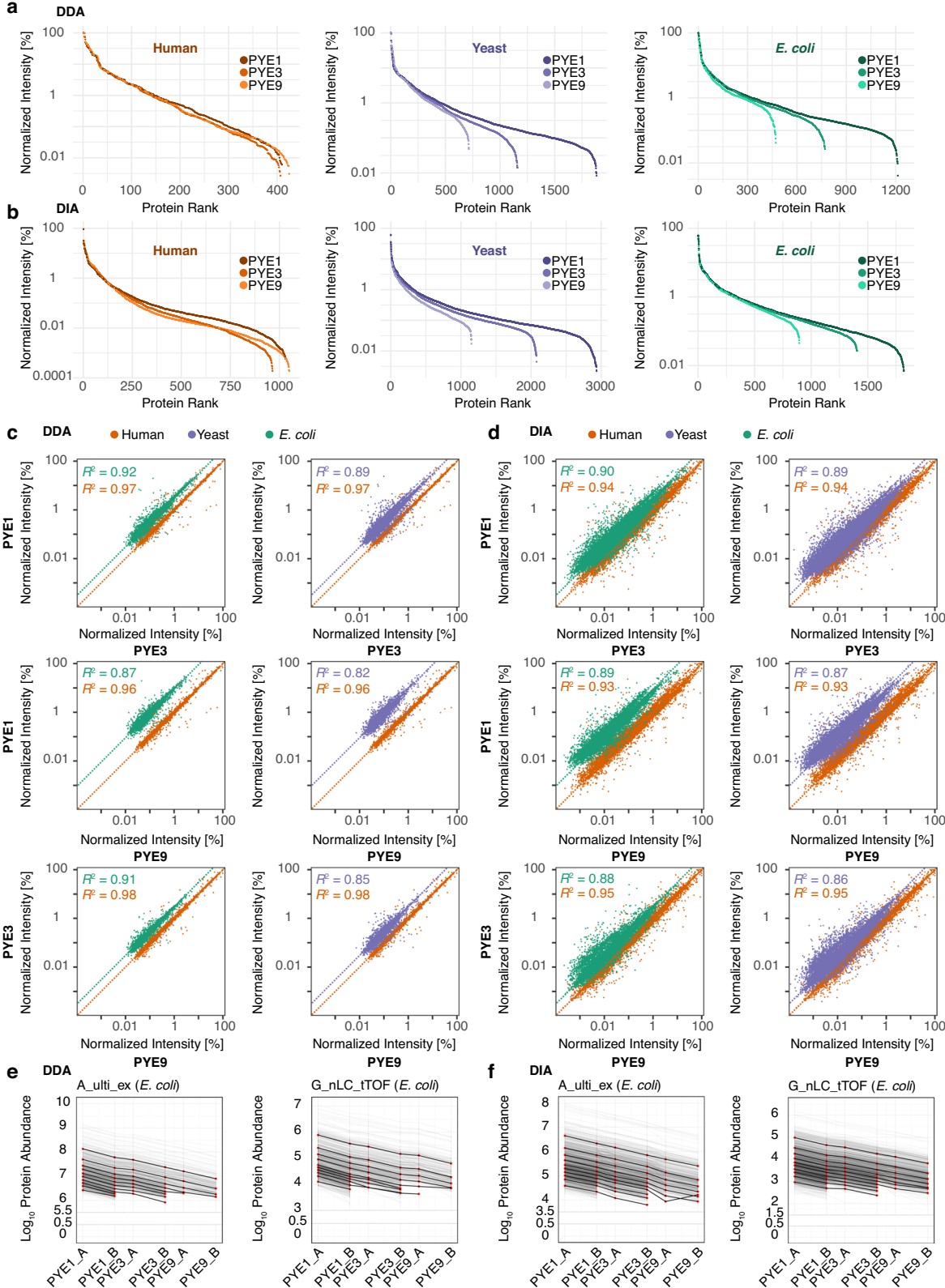

## DIA workflows show superior quantitative performance over DDA-based approaches independent of the LC-MS setup used

As reproducibility is a key aspect in large-scale proteomic studies and we observed a strong influence of the acquisition mode in terms of proteome coverage, we next compared the quantitative performance between the different DIA and DDA datasets in more detail. In terms of run-to-run reproducibility, i.e., reproducibility between replicate

injections, DIA-based LC-MS workflows markedly outperformed the DDA-based methods independent of the LC-MS setup used. Median coefficients of variation (CVs) of protein abundances ranged between 6.4% and 54.7% (average 15.4%) for DDA and between 3.3% and 9.8% (average 5.9%) for DIA analyses as exemplarily shown for PYE1 A in Fig. 4a, b (similar numbers were observed for PYE1 B, Supplementary Fig. 7a,b, Supplementary Data 4). Among the DIA datasets, data

**Fig. 3 | Protein dynamic range and protein intensity distribution across the full PYE sample set integrating data from all sites. a, b** Dynamic range of identified proteins in PYE1, PYE3 and PYE9 across the full dataset (i.e., summarizing normalized protein abundances from all LC-MS runs) split by species and acquisition mode. Panel (**a**) displays the dynamic range for the DDA and panel (**b**) for the DIA dataset. To generate the dynamic range plot, protein intensities were integrated across all different LC-MS setups and divided by the maximum observed intensity, set to 100%. Correlation of normalized protein abundances between PYE1, PYE3 and PYE9 for **c** the DDA and **d** DIA datasets. Protein intensities were averaged and normalized separately for each LC-MS setup to the highest LFQ intensity of each individual setup. Dotted lines indicate the expected values for the comparison between the different PYE dilutions. Coefficient of determination ($R^2$) is displayed in the graphs for human (orange), yeast (violet), and *E. coli*: (green) proteins. Linearity of *E. coli* protein LFQ abundances analysed in DDA (**e**) and DIA mode (**f**), exemplarily depicted for two setups, A_ulti_ex and G_nLC_tTOF. The design of the PYE sample set allows to compare LFQ values of *E. coli* spike-ins across six dilution levels. We binned proteins according to their LFQ abundance values (averaged across six replicate injections) in sample PYE1A into 10 equal-sized bins calculating for each bin the median value (red dot). Median abundance values for these bins (i.e., associated with the proteins assigned to initial bins) were calculated and are plotted for all six PYE samples. Light grey lines represent individual protein response curves. Source data are provided as a Source Data file.

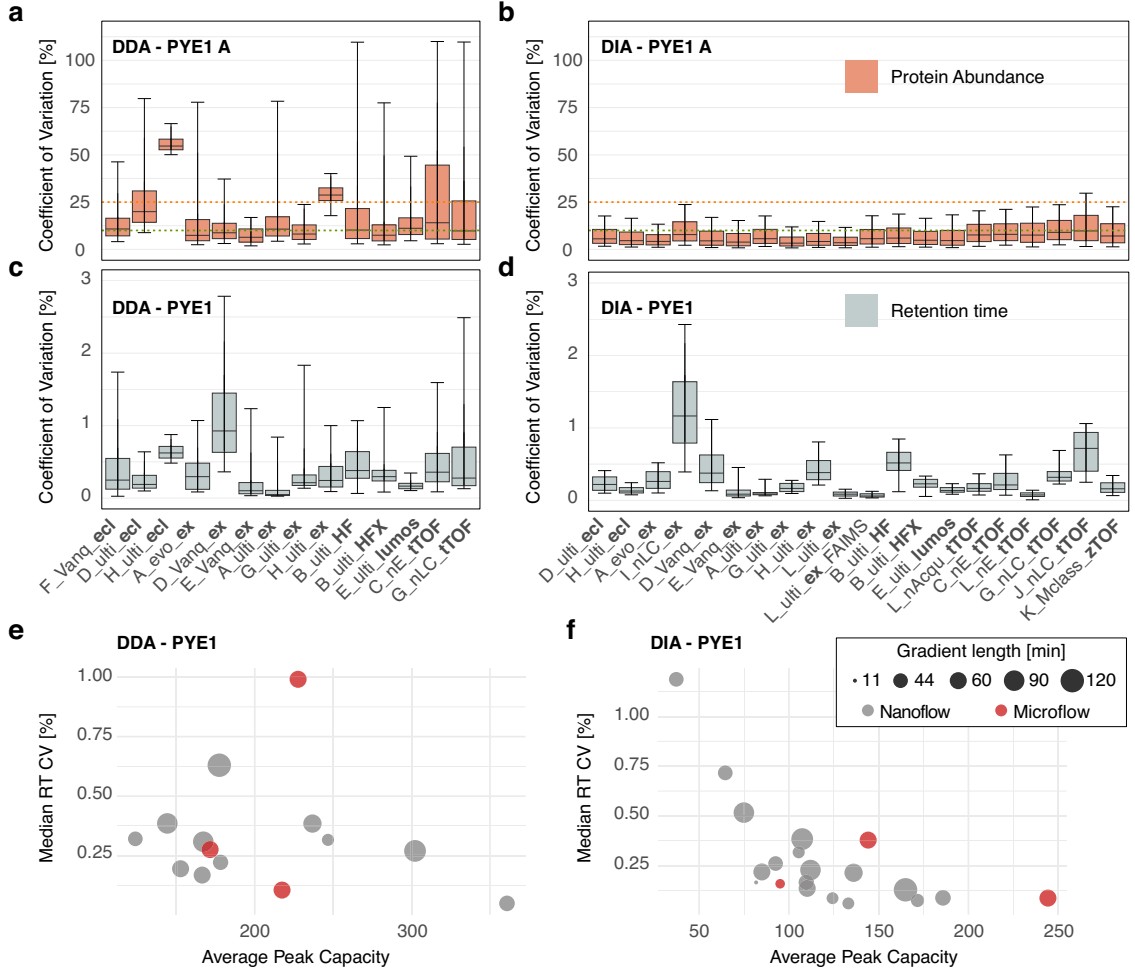

**Fig. 4 | Reproducibility of LC-MS analyses.** Coefficients of variation (CVs) of protein abundances for replicate analyses of sample PYE1 A were calculated for each LC-MS setup revealing lower quantitative reproducibility for **a** DDA as compared to **b** DIA approaches. Boxplot center lines represent the median value, boundaries the interquartile range and whiskers the 5th/90th percentiles of the dataset. The red line marks 25% CV and green line 10% CV. For detailed information on the number of replicate injections for each setup (*n* = 6 in most cases) see Table 1. **c, d** Evaluation of RT stability (displayed as CVs of RT, calculated for sample PYE1, *n* = 12 in most cases, six technical replicates for each, PYE1 A and PYE1 B, for details see Table 1) shows reproducible elution of peptides for most of the LC setups. Center line in the boxplots represents the median value, bounds of boxes the interquartile range and whiskers the 5th/90th percentiles of the dataset. **c** the RT CVs for the DDA and panel (**d**) for the DIA datasets. Median RT CVs are plotted against the chromatographic peak capacity for each chromatographic setup for the **e** DDA and **f** DIA datasets. Dot sizes indicate gradient length. Gray: Nanoflow, red: Microflow setup, see also Supplementary Data 4. Source data are provided as a Source Data file.

derived from timsTOF instruments showed slightly higher variance (average of median CVs: 8.16%) as compared to the other DIA setups (4.87%). Similar trends were also observed for the data processed in FragPipe, where the DIA-based methods display lower CVs as compared to their DDA-based counterparts (Supplementary Fig. 7c, d).

As very different chromatographic setups were used in the present study, including those at higher flow rates (sites D, E, F, and

K), we additionally assessed chromatographic performance evaluating the retention time (RT) stability across replicate runs, an essential factor particularly for label-free quantitative workflows where features are mapped across multiple runs[37]. Overall, the peptide elution behavior was stable and highly reproducible for most of the LC settings, with median RT CVs below 0.35% across all 34 setups (Fig. 4c, d). Only few setups (nine in total) displayed slightly higher RT

variance with median values above 0.35%, including two setups (D_Vanq_ex DDA, I_nLC_ex DIA) with markedly higher RT CVs (0.99% and 1.19%) compared to the other setups. In contrast to our expectations, we observed no marked differences regarding RT CV or peak capacity (Fig. 4e, f) between the micro- and nano-flow settings in the present dataset. We further noted that, independent of gradient length or flow rate, a less reproducible peptide elution, i.e., higher RT CVs, also correlated with an overall lower chromatographic peak capacity (Fig. 4e, f. Supplementary Data 4). This observation was slightly more prevalent for the DIA as compared to the DDA dataset. Particularly DIA methods can benefit from a high peak capacity, i.e., good chromatographic performance, as many downstream processing tools use chromatographic elution profiles for spectral deconvolution and mapping of precursor and product ions.

The present multicenter study comprises 13 matching DDA and DIA datasets, where exactly the same LC-MS setup was used for data acquisition (i.e., analysing the samples at the same site on the same LC-MS platform, with the same LC method and column setup, see Table 1 and Supplementary Data 1), which allows a direct back-to-back comparison of the two acquisition modes (Fig. 5). The majority of these datasets were acquired on Orbitrap platforms. Summarizing the quantitative results of the PYE1 analysis across all 13 LC-MS setups, we found that DIA approaches show on average higher accuracy and precision as compared to the DDA-based methods (Fig. 5a, Supplementary Data 5): The interquartile range (IQR, Q75-Q25) of the global distribution of log-transformed ratios (log$_2$(PYE1 A/PYE1 B)) of protein abundances, averaged across all 13 DIA datasets, ranged between 0.07 for plasma, 0.16 for *E. coli* and 0.22 for yeast proteins. The variance was higher in the case of DDA (IQR$_{plasma}$ = 0.11, IQR$_{E. coli}$ = 0.19 and IQR$_{yeast}$ = 0.27). Moreover, calculated values (averaged across all 13 datasets) were closer to the expected ratios for plasma and for *E. coli* proteins in the DIA runs as compared to the DDA analysis. Only in case of yeast proteins, the DDA measurements showed on average better accuracies as compared to DIA with an absolute difference from the expected ratio of 0.14 versus 0.18 in case of DIA. This effect can most likely be attributed to the higher proteome coverage in DIA, where particularly medium and low-abundant proteins, that are not detected by DDA, can be still identified and quantified (Fig. 5b–e). Overall, similar trends in terms of quantitative precision and accuracy can also be seen for PYE 3 and PYE 9 where in most cases, DIA methods outperform DDA-based approaches, as exemplarily shown for an Orbitrap as well as a timsTOF setup in Fig. 5b, c and Table 2. Interestingly, both timsTOF setups (C_nE_tTOF and G_nE_tTOF) displayed a systematic error of accuracy values in the same direction for both the DDA and DIA dataset.

Additionally, we compared the data completeness for identified yeast proteins across all 13 DDA and DIA datasets. To this end, we mapped the yeast proteins identified in the PYE1 B sample, ranked by their abundance, to those identified in PYE1 A summarizing the results across all 13 datasets. In line with the higher proteome coverage and overlap between the technical replicates (Fig. 2a), the 13 DIA datasets showed a markedly higher data completeness for the yeast spike-in as compared to their matching DDA datasets (Fig. 5d, Supplementary Fig. 8): While the DDA dataset displayed 50% missing values already at protein rank 828, the DIA data reached a value of 50% missingness at protein rank 1637 (Fig. 5d). Additionally, we directly compared the two datasets mapping the yeast proteins identified in sample PYE1 B (Fig. 5e). Here, 50% missing values occurred at protein rank 742, and around 1200 yeast proteins were uniquely detected in the DIA PYE1 B dataset, further highlighting the superior performance of DIA compared to DDA-based methods in the present study.

## Comparison of DIA workflows shows robust quantitative performance for all LC-MS setups and highlights the challenges of accurately quantifying low-abundant proteins

Next, we evaluated the quantitative performance of the 20 different DIA setups. All LC-MS setups demonstrated excellent performance in terms of accuracy and precision for label-free quantification of highly abundant proteins in the PYE sample set (Fig. 6, Supplementary Figs. 9–12). However, for proteins in the low abundant range accurate quantification can still be challenging. Yeast proteins make up the smallest proportion of the PYE samples A and B by quantity. Moreover, yeast proteins are spiked in at a ratio of 1:3, while the ratio for *E. coli* proteins is 1:2, making it even more challenging to estimate the correct ratio between samples A and B for yeast as compared to *E. coli* or human proteins. This is also reflected in the results. For example, variance is markedly higher in the PYE1 set for yeast as compared to *E. coli* proteins (IQR of the global distribution of log$_2$(FC) values across all 20 datasets: IQR$_{yeast}$ = 0.23 and IQR$_{E. coli}$ = 0.17, see also Fig. 6a and Supplementary Data 6). Upon additional dilution of the yeast and *E. coli* proteomes in the PYE3 and the PYE9 samples (Fig. 6b, Supplementary Fig. 12), variance increases for both species (to IQR$_{yeast}$ = 0.27 and IQR$_{E. coli}$ = 0.19 in the PYE9 set). Interestingly, precision slightly improves for human proteins from PYE1 to PYE9, likely due to a decrease of the yeast and *E.coli* proteome background. Particularly in the lowest abundance tertile accurate and precise quantification still remains challenging. This becomes evident when looking exclusively at the log$_2$(FC) distributions of the proteins in the low abundance range (i.e., the tertile of the dataset encompassing the proteins with the lowest abundance values, Fig. 6c, d). Across all dilutions, comprising sample sets PYE1 to 9, accuracy and precision are markedly lower, particularly for *E.coli* and human proteins, in the lowest abundance tertile as compared to the full dataset that includes also the mid and high abundant proteins (Fig. 6a, b Supplementary Fig. 12).

Looking at the full PYE dataset (Fig. 6e), accuracy follows a similar trend as the precision. Averaging across all datasets, accuracies of calculated log$_2$(FC) values for human proteins improved from the PYE1 to the PYE9 sample set (with an average absolute difference between median and expected values of 0.10 in PYE1 and 0.01 in PYE9; Fig. 6a, b, Supplementary Data 6). Comparing yeast and *E. coli* proteomes, deviations from the expected ratios are markedly higher for yeast as compared to *E. coli* proteins in all sample sets, i.e., PYE1, PYE3 and PYE9 (Fig. 6e). Accuracy is similar for yeast proteins between samples PYE1, PYE3 and PYE 9, whereas there is a slightly higher deviation from the expected values in PYE9 as compared to PYE1 for *E. coli* proteins.

Interestingly, most TOF setups show a similar trend regarding their LFQ values, which display a consistent shift from the expected values for yeast and human proteins in the same direction (Fig. 6a, b, e), indicating a potential issue with background correction for the TOF data overestimating LFQ abundances for low abundant proteins[7]. This effect can potentially be attributed to an overall higher background in TOF mass spectra as compared to those derived from Orbitrap platforms, or alternatively to different background subtraction algorithms. For the Orbitrap LC-MS setups, we observe varying effects. For example, the two micro-flow setups (D_Vanq_ex and E_Vanq_ex), show the highest accuracy and precision for human proteins as compared to all other setups. However, deviations from the expected log$_2$(FC) values point to an underestimation of LFQ values for low-abundant yeast and *E. coli* proteins. For other Orbitrap setups, e.g., D_ulti_ecl, H_ulti_ecl, H_ulti_ex, we observe a systematic error (in PYE1 and PYE3) of the calculated log$_2$(FC) values for all species towards a higher log$_2$(FC) than expected.

To better understand some effects, we additionally evaluated for the yeast proteins if some metrics, such as data points per peak, number of identified proteins, peak capacity, or mean CV, correlate with quantification accuracy and precision at a proteome-wide scale

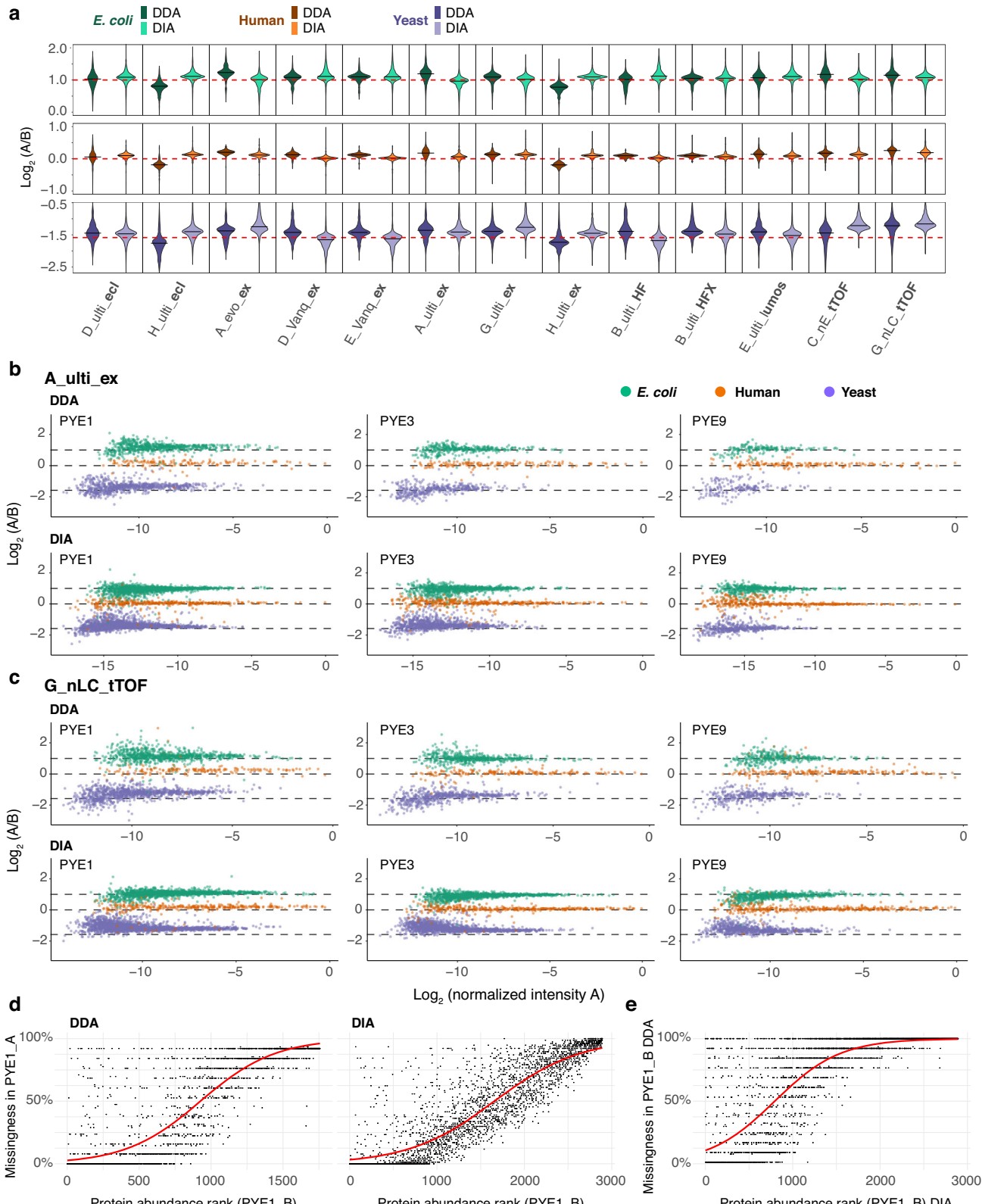

**Fig. 5 | Lower number of missing values and better quantitative performance of DIA- as compared to DDA-based methods. a** Violin plots of log-transformed ratios (log$_2$(PYE1 A/PYE1 B)) of protein abundances for matching DDA and DIA LC-MS setups. Solid lines within the violin plot indicate the median log$_2$(A/B) value for each setup and red dashed lines the expected log$_2$(A/B) values for human (orange), yeast (violet), and *E. coli* (green) proteins (Supplementary Data 5). Log-transformed ratios (log$_2$(A/B)) of proteins were plotted over the log-transformed intensity of sample A for DDA and DIA data acquired with the same LC-MS setup on **b** an Orbitrap as well as **c** a timsTOF platform. **d** Percentage of missing values for yeast proteins in PYE1 A as compared to PYE1 B (ranked by protein abundance) for the DDA and DIA dataset. **e** Percentage of missing values for yeast proteins in the PYE1 B DDA dataset as compared to the PYE1 B DIA dataset dependent on protein abundance across all 13 LC-MS setups displayed in panel (**a**). **d**, **e** X-axis: Rank as defined by the average normalized intensity (INT$_{Protein}$/INT$_{max}$) across all 13 setups. Y-axis: Missingness (1-(number of detections/number of runs)) across all 13 setups and injection replicates as percent values.

**Table 2 | Metric summary for the datasets shown in Fig. 5**

| Species | DDA | | | DIA | | | Sample |
|---|---|---|---|---|---|---|---|
| | IDs | Accuracy[a] | Precision[b] | IDs | Accuracy[a] | Precision[b] | |
| Lab A ulti_ex | | | | | | | |
| *E.coli* | 833 | 0.19 | 0.24 | 1492 | −0.04 | 0.17 | PYE1 |
| Human | 175 | 0.19 | 0.19 | 399 | 0.06 | 0.10 | |
| Yeast | 982 | 0.23 | 0.27 | 2183 | 0.17 | 0.18 | |
| *E.coli* | 481 | 0.05 | 0.17 | 1061 | −0.03 | 0.19 | PYE3 |
| Human | 186 | 0.09 | 0.08 | 466 | 0.08 | 0.14 | |
| Yeast | 569 | 0.09 | 0.29 | 1411 | 0.21 | 0.22 | |
| *E.coli* | 217 | 0.09 | 0.19 | 591 | −0.02 | 0.16 | PYE9 |
| Human | 187 | 0.10 | 0.11 | 460 | 0.00 | 0.15 | |
| Yeast | 240 | 0.15 | 0.38 | 704 | 0.02 | 0.21 | |
| Lab G nLC_tTOF | | | | | | | |
| *E.coli* | 1095 | 0.15 | 0.22 | 1614 | 0.07 | 0.14 | PYE1 |
| Human | 246 | 0.26 | 0.16 | 368 | 0.19 | 0.12 | |
| Yeast | 1548 | 0.37 | 0.32 | 2530 | 0.42 | 0.22 | |
| *E.coli* | 713 | −0.02 | 0.18 | 1218 | −0.07 | 0.14 | PYE3 |
| Human | 226 | 0.07 | 0.09 | 393 | 0.07 | 0.09 | |
| Yeast | 904 | 0.17 | 0.33 | 1742 | 0.36 | 0.26 | |
| *E.coli* | 456 | 0.02 | 0.24 | 749 | −0.09 | 0.18 | PYE9 |
| Human | 279 | 0.11 | 0.13 | 412 | 0.06 | 0.14 | |
| Yeast | 565 | 0.23 | 0.29 | 900 | 0.29 | 0.24 | |

The table summarizes number of identified protein groups, median accuracy and precision (Q075-Q025) for sample sets PYE1 to PYE9 analysed in DDA and DIA mode on an Orbitrap (site A) and a timsTOF setup (site G). Full data across all sites is found in Supplementary Data 5.
[a]Accuracy: deviation of the experimental log-transformed ratio ($\log_2(A/B)$) of protein abundances from the expected value, Q050.
[b]Precision: Q075–Q025.

(exemplarily shown for PYE1, Fig. 6f and Supplementary Fig. 13, Supplementary Data 7) and found that the median deviation from expected values slightly increased in datasets with higher ID numbers. Moreover, in datasets that display higher accuracies, more data points were recorded across a chromatographic peak. Interestingly, we observed a slightly opposing trend regarding the precision (Supplementary Fig. 13), which improved when higher numbers of proteins were identified. Other factors, i.e., mean CV or data points per peak, did not correlate with improved precision, i.e., lower variance.

**Interlaboratory LC-MS analyses employing identical setups and instrumental parameters demonstrate robust method transferability**

To leverage the advantages of multicenter studies, particularly regarding method transferability and interlaboratory reproducibility, we re-analysed the PYE1 sample set at site L using the Ultimate/Exploris DIA configurations from sites G and H (G_ulti_ex, H_ulti_ex), as well as the EASY-nLC 1200/timsTOF DIA setup from site G (G_nLC_TOF). Re-analysis of the PYE1 sample at site L, using the LC-MS configurations from the original sites, yielded highly comparable numbers of protein and peptide identifications (Fig. 7a) with substantial overlap (Fig. 7b), effectively demonstrating the interlaboratory transferability of the methods. Additionally, the quantitative profiles closely mirrored the distribution patterns observed in the original round robin dataset (Fig. 7c). Of note, the H_ulti_ex and G_nLC_tTOF DIA setups from sites G and H yielded the highest proteome coverage in the round robin study. In line with the round robin data, remeasurements at site L also provided lower proteome coverage for the G_ulti_ex setup, which uses the same LC-MS and column setup as H_ulti_ex, but half the analysis time, i.e., 60 min versus 120 min and slightly different DIA method with adapted lower cycle time (see Supplementary Data 1).

To further explore how the number of IDs is influenced on a distinct platform, we additionally conducted a back-to-back comparison of the timsTOF setups from sites G and L. G_nLC_TOF, the timsTOF setup providing the highest IDs, uses an IonOpticks Aurora column (75 μm ID × 25 cm) for peptide separation running a 30 min gradient at 300 nL/min (Fig. 7d, e). We analysed the PYE1 sample using the MS method of site G but the LC setting from site L (Bruker PepSep setup, 150 μm ID × 25 cm, 35.5 min gradient at 850 nL/min). This resulted in a marked drop in the number of identified proteins and peptides (Fig. 7d, e). In contrast, we observed no marked differences in IDs between the MS methods from sites G and L (30 Da versus 25 Da fixed window schemes, different IMS range and cycle time). These data clearly indicate, that the LC setup used by site G (IonOpticks Aurora column 75 μm ID × 25 cm, 30 min gradient at 300 nL/min, final amount of 28% (v/v) ACN) outperforms the conditions used by site L in the round robin study (Bruker PepSep column 150 μm ID × 25 cm, 35.5 min gradient at 850 nL/min up to 38% (v/v) ACN).

To evaluate whether the findings from the PYE analyses are applicable to native plasma samples, we analysed a neat plasma sample without any spike-ins across three different sites using four different Orbitrap-based LC-MS setups from the round robin study (Fig. 7f, g). Consistent with the results obtained from the PYE analyses, we observed similar trends regarding the number of IDs as compared to the round robin study with a high degree of overlap (Fig. 7g). In line with the round robin study, the setup with the longest gradient and analysis time, i.e., H_ulti_ex, provided the best proteome coverage also for the neat plasma sample. This clearly demonstrates that depending on the scope of a (clinical) study one has to balance proteome depth, quantitative performance, and sample throughput when choosing an LC-MS setup for plasma analysis.

## Discussion

Over the past two decades, plasma proteomics has evolved significantly, progressing from basic protein cataloguing to sophisticated workflows that quantify thousands of proteins with high

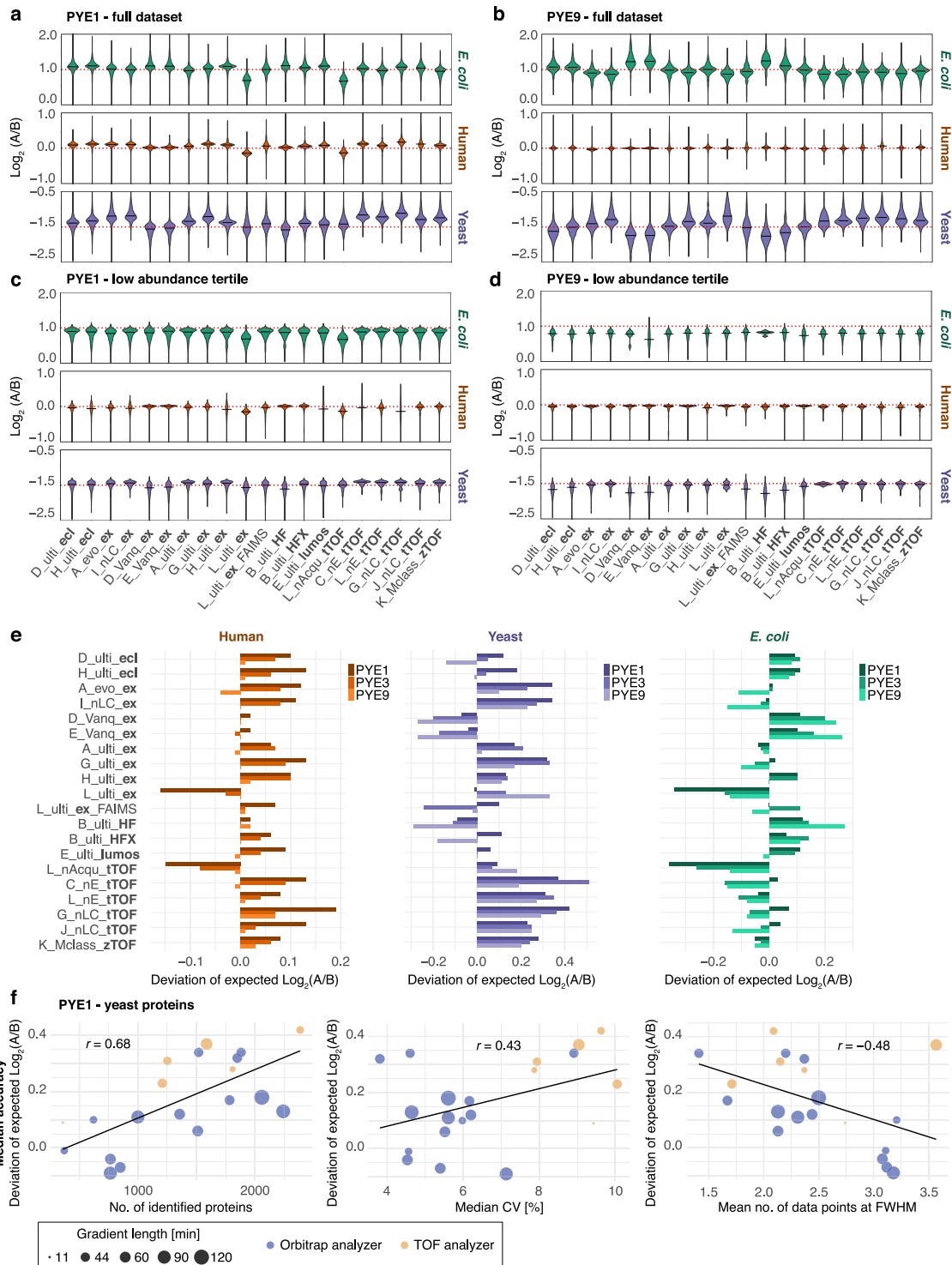

**Fig. 6 | Quantitative metrics of the DIA dataset acquired with 20 different LC-MS setups.** Violin plots of log-transformed ratios (log₂(A/B)) of protein abundances for **a** the full PYE1 and **b** PYE9 set (Supplementary Data 6). Violin plots of log-transformed ratios (log₂(A/B)) of protein abundances in the lowest intensity tertile for **c** the PYE1 and **d** PYE9 set. Solid black lines within the violin plot indicate the median log₂(A/B) value for each setup and red dashed lines the expected log₂(A/B) values for human (orange), yeast (violet), and *E. coli* (green) proteins. **e** Deviation of the median log-transformed ratio (log₂(A/B)) of protein abundances

from the expected value. Plots summarize data for human (orange), yeast (violet) and *E. coli* proteins (green) for the PYE1, PYE3 and PYE 9 datasets. **f** Correlation of median accuracies (calculated for yeast proteins in the PYE1 set) with other metrics such as number of identified proteins (left), median CV [%] of protein abundances (middle) and number of datapoints acquired across the chromatographic peak (right panel). Dot sizes indicate gradient length. Blue: Orbitrap, orange: TOF analyzer, see also Supplementary Data 7. Source data are provided as a Source Data file.

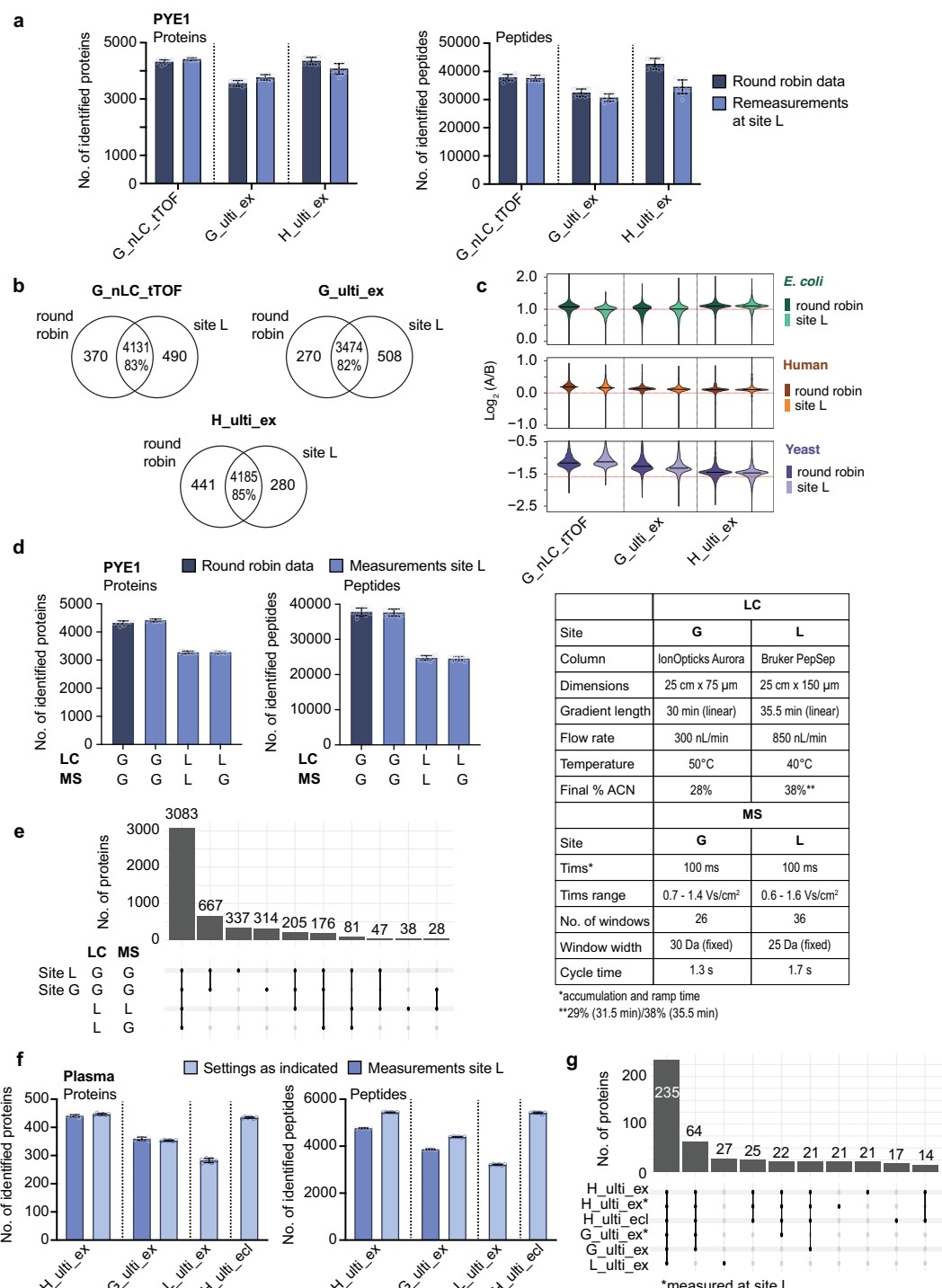

**Fig. 7 | Reproducibility and inter-laboratory transferability of methods.**
**a** Numbers of identified proteins and peptides in the PYE1 samples comparing data from the round robin study with remeasurement using the same setup as indicated in site L ($n$ = 12, six technical replicates for each, PYE1 A and PYE1 B). **b** Overlap of identified protein groups between the round robin data and the remeasurements in site L. **c** Violin plots of log-transformed ratios (log$_2$(A/B)) of protein abundances in PYE1 for the round robin data and the remeasurements at site L. **d** Numbers of identified proteins and peptides in the PYE1 samples comparing the nLC_tTOF round robin data from site G with remeasurements in site L using different LC and MS settings (as indicated in the table on the right; $n$ = 12, six technical replicates for

each, PYE1 A and PYE1 B). **e** Upset plot depicting the overlap of identified protein groups from the measurements in (**c**). **f** Numbers of identified proteins and peptides in neat plasma samples (without spike-ins, $n$ = 6 replicate injections) analysed at three different sites using LC-MS setups as indicated (for more details see Supplementary Data 1). LC-MS setups G_ulti_ex and H_ulti_ex were used at the respective sites as indicated (light blue) as well as at site L (darker blue). **g** Upset plot depicting the overlap of identified protein groups from the measurements in (**g**). Source data are provided as a Source Data file. In panels (**a, d, f**) points represent individual injections; bars and error bars show mean ± sd.

precision[16,38,39]. Despite these advancements, plasma remains a challenging sample matrix for LC-MS-based proteomics due to its tremendous dynamic range[3,4]. High-abundant proteins, such as albumin and immunoglobulins, can overshadow lower-abundance proteins, many of which hold potential as biomarkers for disease. Early plasma proteomics studies using DDA-based methods identified typically only a few hundred proteins[3,40], with a bias toward high-abundant ions and inconsistent detection of low-abundance peptides across analyses. Workflows incorporating off-line fractionation and depletion strategies improved proteomic depth, extending coverage to over 1000 proteins identified per sample, albeit with significant time costs[10,11]. DIA-based approaches address challenges of dynamic range and reproducibility by capturing all ions in a mass-to-charge range without bias[41], thereby improving consistent and reproducible detection of low-abundance proteins. Coupled with high-resolution MS, DIA enables robust, efficient identification of over 500–1000 proteins from neat plasma, minimizing fractionation needs and advancing biomarker discovery in large-scale studies[42,43]. While some studies show DIA outperforms DDA in plasma proteomics by capturing a broader ion range and enhancing low-abundance protein quantification[44], systematic comparisons across various LC-MS platforms are limited. Such research is essential, as differences in LC and mass spectrometer hardware configurations affect resolution, sensitivity, and scan speed, impacting DIA and DDA performance. Additionally, variations in LC parameters, including gradient length, column and flow rate, also influence peptide separation and detection[45,46]. Despite the high potential of LC-MS proteomics for protein identification and quantification, its diagnostic use is limited by a lack of standardized workflows and validation processes required for accreditation[4,47]. Cross-platform studies would clarify how different parameters affect DDA and DIA, guiding method selection for standardization and demonstrating each method's practical benefits across diverse workflows for plasma proteomics.

Here, we designed and conducted a multicenter study including twelve partner sites of the German research cores for mass spectrometry in systems medicine (MSCoreSys) to assess label-free quantification performance on a benchmark sample set, simulating the high protein dynamic range typical of neat plasma. Including multiple sites and a diverse range of LC-MS setups, with data centrally analysed using standardized software (MaxQuant for DDA and DIA-NN for DIA, FragPipe for both acquisition modes), lends robustness to our findings. We focused on critical parameters such as intra- and inter-laboratory reproducibility, highlighting proteins consistently detected across LC-MS platforms at various sites. Additionally, we evaluated the total number of quantified proteins, quantitative reproducibility, data completeness, and the precision and accuracy of quantification.

Unlike previous benchmark studies that used a HeLa digest as a matrix[7,19], we generated a multispecies sample set based on a human tryptic plasma digest with varying spike-in amounts of tryptic digests of yeast and *E. coli* proteomes. This effectively simulates the high protein dynamic range of human plasma and the low abundance of potential biomarker candidates[4,22]. Specifically, the initial sample set (PYE1 A/B) was diluted incrementally at a 1:3 ratio with a human tryptic plasma digest, reaching maximum dilution in PYE9 A/B, where human plasma proteins constituted 98.9% of the total protein mass, with yeast and *E. coli* proteins comprising the remaining 1.1%. Notably, even at these low spike-in levels, current-generation instrument platforms provided precise and accurate label-free quantification of several hundreds of yeast and *E. coli* proteins in the present study. Our analysis of proteome coverage across various LC-MS setups, acquisition modes, and PYE sample dilutions showed that DIA consistently outperformed DDA in protein and peptide ID numbers, with DIA workflows offering greater run-to-run reproducibility and higher consistency in protein identification. Notably, the detection of hundreds of non-human proteins across the full dynamic range indicates

that current DIA based proteomic platforms are likely to cover the entire plasma proteome in the upper 3–4 orders of magnitude of dynamic range. Compared to DDA, DIA-based workflows achieved up to eight times higher proteome coverage, improved quantitative reproducibility, and significantly fewer missing values, consistent with previous studies[24,41,48]. However, identifications on the protein as well as peptide level can be significantly impacted by the software tool and settings used for data processing and database search. The gap in proteome coverage between the DDA and DIA dataset markedly decreased upon data processing in FragPipe highlighting the importance of exploring different software tools and parameters for data analysis when planning a (clinical) study. Overall, our data demonstrate that a technical reproducibility between replicates with less than 6% CV are achievable across different setups and instrument platforms using DIA-based approaches. This indicates that precise label-free quantification is feasible even in a complex matrix such as plasma using state-of-the-art workflows. This high precision and accuracy in label-free quantification underscore DIA as the preferred acquisition method for the analysis of plasma and other high-dynamic range proteomes using LC-MS. Interestingly, while DIA excelled in identification and quantification metrics, our study also revealed that longer gradient times generally led to higher ID rates. However, differences in the LC-MS setup including, for example, instrument type, column characteristics, etc., more profoundly affected detection rates, even with similar gradient durations. Notably, all participating sites used chromatographic setups that were optimized for plasma proteomics to provide optimal sensitivity, reproducibility, and data quality. Optimizing chromatography is thought to be particularly important in DIA due to its continuous, wide-window sampling, where optimal peak sharpness and separation are essential for capturing high-quality fragment ion spectra and maximizing identification rates. However, in contrast to our expectations, we did not observe a significant correlation of chromatographic parameters, i.e., peak capacity or retention time stability, with the respective proteomic coverage or quantitative metrics. This may likely be attributable to the multiparametric setup of the participating labs and the high dynamic range of the PYE sample set.

Although challenges remain in accurately quantifying low-abundance proteins in plasma proteomics, our findings underscore the significant improvements in LC-MS-based workflows in recent years, which now offer enhanced quantitation accuracy and precision. Here, our findings align with a recent study in which a mixed proteome benchmark set based on HeLa digest was used to assess the impact of DIA-NN processing parameters on the evaluation of QE-HF data and a cross-platform comparison. In the mentioned study, a CV cut-off of 5% was suggested as a threshold for deeming workflows or datasets quantitatively reproducible[29]. Looking ahead, we anticipate that further developments in chromatography and mass spectrometric instrumentation will push the boundaries of both proteome depth and data quality. While reference studies from the early 2000s demonstrated state-of-the-art plasma proteomics with the identification of around 100–200 proteins, it is now routinely possible to achieve a coverage of >500–1000 proteins[42,43]. Recent comparisons between instruments, like the Orbitrap Exploris 480 and Astral, demonstrate promising gains in sensitivity, highlighting the potential for even greater precision in low-abundance protein quantification[30], particularly also with respect to plasma analysis[49].

Our dataset not only identifies areas for further improvement but also serves as a valuable resource for software development, offering a comprehensive overview of current technological capabilities in LC-MS workflows. Moreover, we could demonstrate how multicenter studies can facilitate the reproducible transfer of methods across different sites. These advancements show how LC-MS technology has evolved into a robust and reliable platform with great potential for biomarker discovery and validation. It sets the stage for a continuously

increasing role of quantitative proteomics in systems medicine and clinical research.

## Methods

### Reagents and chemicals

Unless otherwise stated, all solvents (HPLC and Ultra LC-MS grade) were purchased from Roth and all chemicals were obtained from Sigma.

### Preparation of the PYE benchmark sample set

Human plasma was commercially obtained from BioCat GmbH (Heidelberg, Germany) and tested negative for HIV, ZIKA Virus, STS (Syphilis) and Hepatitis B/C. A pure culture of the *Saccharomyces cerevisiae bayanus*, strain Lalvin EC-1118 was obtained from Eaton (www.eaton.com). *E. coli* was purchased from Thermo Fisher Scientific.

*E. coli* cells were lysed using a urea-based lysis buffer (7 M urea, 2 M thiourea, 5 mM dithiothreitol (DTT), 2% (w/v) CHAPS). Lysis was further promoted by sonication at 4 °C for 15 min using a Bioruptor (Diagenode, Liège, Belgium). Yeast proteins were extracted using alkaline pre-incubation with 0.1 M NaOH (VWR, USA) followed by an additional incubation step in lysis buffer containing 1% (w/v) SDS (Carl Roth, Germany) at 95 °C.

After lysis, the concentrations of *E. coli* and yeast proteins were determined using the Pierce 660 nm protein assay (Thermo Fisher Scientific) according to the manufacturer´s protocol. Neat plasma was diluted 166-fold in urea-based buffer (7 M urea, 2 M thiourea, 5 mM dithiothreitol (DTT), 2% (w/v) CHAPS) prior to digestion.

Human plasma, yeast and *E. coli* proteins were digested on an Biomek i7 robotic pipetting platform (Beckman Coulter Life Sciences, Indianapolis, USA) equipped with a positive pressure adapter (Amplius, Germany) using an adapted filter-aided sample preparation (FASP) protocol[50]. All digestion steps are detailed in Distler et al.[51] and were implemented on the Biomek i7 liquid-handling robot. Unless stated otherwise, each step of the semi-automated FASP workflow was performed as described[51] and carried out by the liquid-handling robot applying a positive pressure of 500 mbar for 6–15 min to force the liquid through the filter membranes. All volumes were adapted to 100 µL/well, except for the trypsin digestion and the elution steps after overnight digestion: Sample aliquots (corresponding to 30 µg of protein per well) were manually transferred onto AcroPrep Advance 96-well 350 µL 30 K Omega filter plates (Pall Cooperation, USA) which had been additionally preconditioned with 0.1% (v/v) formic acid (FA) and urea-based lysis buffer (7 M urea, 2 M thiourea, 5 mM dithiothreitol (DTT), 2% (w/v) CHAPS) in case of plasma and *E. coli*. After sample transfer, membranes were washed once with a urea-based wash buffer (8 M urea, 0.1 M Tris-HCl, pH 8.5). Proteins were then reduced for 15 min at 56 °C using 8 mM DTT dissolved in the urea-based wash buffer followed by an additional washing step. Afterwards, proteins were alkylated with 50 mM iodoacetamide (IAA, in urea-based wash buffer) for 20 min at room temperature. Excess IAA was removed by two washes using the urea-based wash buffer and additionally quenched with 8 mM DDT for 15 min at 56 °C. Afterwards, the membrane washed twice with urea-based wash buffer followed by three additional washing steps with 50 mM NH₄HCO₃. Proteins were then digested overnight at 37 °C adding 40 µL of trypsin (Trypsin Gold, Promega, Madison, WI) dissolved in 50 mM NH₄HCO₃, 0.02% (w/v) DDM in water at an enzyme-to-protein ratio of 1:50 (w/w) corresponding to 0.6 µg of trypsin per well. After digestion, tryptic peptides were recovered from the membrane adding 40 µL 50 mM NH₄HCO₃. Flow-throughs were acidified with FA to a final concentration of 0.1% (v/v) FA. Tryptic peptides from multiple well plates were pooled in case of all three species to obtain digest stock solutions for the generation of the PYE sample set.

Digest quality of the different stocks was assessed by LC-MS (checking for impurities, peptide abundances, total ion current as well as number of peptide and protein IDs). Tryptic peptides were subsequently mixed in predefined ratios to generate hybrid proteome samples. In total, the PYE benchmark set comprises six samples, PYE1 A and B, PYE3 A and B, PYE9 A and B (at 2 µg/µL protein). For the PYE1 sample set, tryptic peptides were combined in the following ratios: sample A was composed of 90% w/w human, 2% w/w yeast, and 8% w/w *E. coli* proteins. Sample B was composed of 90% w/w human, 6% w/w yeast, and 4% w/w *E. coli* proteins (Fig. 1a). To generate the PYE3 sample set, samples PYE1 A and B were further mixed with tryptic human plasma peptides at a ratio of 1:3. PYE3 samples were then further diluted threefold with human plasma peptides resulting in the PYE9 sample set.

Afterwards, samples were shipped to all participating sites on dry ice. Shipped sample amounts (i.e., volumes) were dependent on the LC-MS setup used at the respective site providing higher sample amounts to the sites that used a microflow LC-MS setup (see Table 1 and Supplementary Data 1).

### Filter-aided sample preparation (FASP) of neat plasma sample

Blood samples were collected from five healthy volunteers from site L (see also ethics statement). EDTA plasma was prepared by centrifugation at 1780 × *g* for 10 min. The resulting plasma samples were pooled and stored at −80 °C until further processing. Proteolytic digestion of the collected plasma pool was performed using an adapted FASP protocol[50]. All digestion steps are detailed in Distler et al.[51] and were performed manually in a 96-well format analogue to the procedure described above (preparation of the PYE benchmark sample set). In brief, 20 µg of sample material were manually transferred into each well of an AcroPrep Advance 96-well 350 µL 30 K Omega filter plate (Pall Cooperation, USA), which had been preconditioned with 0.1% (v/v) FA.

All volumes, except the volume of the trypsin solution and the steps on day two, corresponded to 100 µL/well. After sample transfer, membranes were washed once with a urea-based wash buffer (8 M urea, 0.1 M Tris-HCl, pH 8.5) followed by reduction of proteins using 8 mM DTT. After two washing steps with urea-based wash buffer, proteins were alkylated with 50 mM IAA. Excess IAA was removed by two washes and quenched with 8 mM DDT. Afterwards, the membrane was washed twice with urea-based wash buffer followed by three additional washing steps with 50 mM NH₄HCO₃. Proteins were subsequently digested overnight at 37 °C with trypsin gold (0.4 µg/well, Promega, USA) in 40 µL 50 mM NH₄HCO₃. After digestion 40 µL 50 mM NH₄HCO₃ were added to the samples to recover tryptic peptides. Samples were acidified with 10 µL 1 % formic acid, which was added to the wells of the 96-well collection plate containing eluted peptides (Waters, USA). Peptides were pooled into one sample pool, which was aliquoted, and lyophilized. Lyophilized sample was sent out to three different partner sites (i.e., sites G, H, and L). At the different sites samples were re-constituted in 0.1% FA (v/v) in water (final concentration of 1 µg/µL) followed by a further dilution to 200 ng/µL in 0.1% FA (v/v) for LC-MS measurements.

### Liquid-chromatography mass spectrometry (LC-MS)

All participating sites were asked to analyse the PYE benchmark sample set using their preferred LC-MS setup for the characterization of plasma samples according to the following measurement scheme: (1) blank injection, (2) Hela QC (e.g., Pierce™ HeLa, Thermo Scientific), (3) two blank injections, (4) PYE samples in the following order, PYE A9, PYE B9, PYE A3, PYE B3, PYE A1, PYE B1), (5) blank injection. All samples had to be analysed in multiple replicates (ranging from three to optimally six replicate injections). No other restrictions were imposed on the study centers regarding LC-MS setup, gradient length, on-column load, etc. Detailed description of the LC-MS settings are provided in

the supplementary section (see Extended Material and Methods section of the Supplementary Info file).

## Raw data processing and label-free quantification

All MS raw data sets of the participating partner sites were collected and centrally analysed in the Tenzer laboratory.

The analysis of DDA data sets was performed using MaxQuant (version 2.3.1.0)[34,35]. Data were searched against a customized database, which was generated by compiling the SwissProt database entries of the human, yeast and *E. coli* reference proteomes and a list of common contaminants (UniProtKB release 2020_03, total of 31,039 entries). For each LC-MS setup and PYE dilution, i.e., PYE1, PYE3 and PYE9, data processing was performed separately. Default MaxQuant parameters were applied, including label-free quantification and match between runs (MBR) enabled. The LFQ minimum ratio count was set to two peptides. Trypsin was chosen as the enzyme and up to two missed cleavages were allowed. Carbamidomethylation of cysteine was set as a fixed modification, while methionine oxidation was specified as variable modification. The FDR was set to 1% for both PSMs and protein level (for parameter file, see Supplementary Data 8).

The DIA data were all processed using DIA-NN (version 1.8.1)[36] applying the default parameters for library-free database search (see Supplementary Data 8). For each LC-MS setup and PYE dilution, i.e., PYE1, PYE3 and PYE9, analysis was performed separately. Data were queried against the same database as the DDA datasets (see previous paragraph). For peptide identification and in-silico library generation, trypsin was set as protease allowing one missed cleavage. Carbamidomethylation was set as fixed modification and the maximum number of variable modifications was set to zero. The peptide length ranged between 7 and 30 amino acids. The precursor $m/z$ range was set to 300–1800, and the product ion $m/z$ range to 200–1800. As quantification strategy we applied the robust LC (high precision) mode with RT-dependent median-based cross-run normalization enabled. We used the build-in algorithm of DIA-NN to automatically optimize MS2 and MS1 mass accuracies and scan window size. Peptide precursor FDRs were controlled below 1%.

PYE data were additionally processed using FragPipe[52] (version 23.0), separately for each LC-MS setup and measurement mode. ZenoTOF raw files were converted to mzML beforehand using MSConvert[53] (version 3.0.20280) with vendor peak picking. The data were searched against the same protein sequence database used for MaxQuant and DIA-NN analyses including the same number of reversed decoy sequences generated by FragPipe. For all DDA experiments the LFQ-MBR workflow was employed, which uses IonQuant[54] for MS1-level quantification. As part of this workflow, normalization of intensities across runs was disabled as we observed some strange effects in the DDA set using cross-run normalisation. For diaPASEF data the DIA_SpecLib_Quant_diaPASEF workflow was used which applies diaTracer[55] for spectrum deconvolution prior to searching. All other DIA experiments were processed using the DIA_SpecLib_Quant workflow, leveraging MSFragger-DIA[31] for direct peptide identification. DIA quantification was performed using the integrated DIA-NN (version 1.8.2 beta 8) module with cross-run normalization disabled via the --no-norm command. To ensure a fair comparison across workflows, key parameters were standardized: the precursor mass tolerance was set from 20 to 20 ppm, and the fragment mass tolerance was 20 ppm. A maximum of one missed tryptic cleavage and one methionine oxidation was allowed. FDR filtering and report generation were conducted using the --picked and --prot 0.01 flags. Default settings were maintained for all other parameters.

## Downstream analysis of PYE data sets

The software reports of each data set (PYE dilution, site and instrument setup) were processed separately. All downstream analyses were conducted after removing reversed sequences and potential contaminants, allowing only proteins identified by 2 or more peptides. In case of the DIA data (DIA-NN), Q.Value, PG.Q.Value, Lib.Q.Value, and Lib.PG.Q.Value had to be additionally below or equal 0.01 for all plots containing quantitative information. For the generation of plots that contain statistics related to the calculated $\log_2$(FC) values between samples A and B (e.g., violin plots, $\log_2$(FC) plots, etc.), proteins had to be identified and quantified in at least three technical replicates in each condition, i.e., sample A and B (for both, DDA and DIA datasets). Of note, Peptides shared between species were excluded for $\log_2$(FC) plots (and violin plots), but taken into account to calculate numbers of identified proteins and peptides. A comprehensive overview of identified and quantified proteins and peptides across all sites for the DDA (MaxQuant) and DIA (DIA-NN) analyses can be also assessed via Zenodo at [https://doi.org/10.5281/zenodo.17131745]. Additionally, an overview of the search results from all software tools uploaded to jPOST/ProteomeXchange (JPST003358/PXD056598) is provided in Supplementary Data 9.

Downstream analysis of the result files from MaxQuant, DIA-NN and FragPipe was performed in R (version 4.3.2)[56] using in-house scripts to calculate and report a set of metrics including the visualization of $\log_2$(FC) changes, identification rates (number of identified proteins and peptides for benchmark species), technical variance (the median CV for protein abundances and retention times), global accuracy (the median deviation of $\log_2$ ratios to the expected value), global precision of quantification (defined by the interquartile range and the standard deviation of $\log_2$ ratios). Identification completeness (bar plots) as well as RT CV plots summarizing results across multiple data sets were inspired by the mpwR ([https://CRAN.R-project.org/package=mpwR](https://CRAN.R-project.org/package=mpwR))[57] and the $\log_2$(FC) plots for individual setups by the LFQBench package[7]. ggplot2 was used to design the plots, except for the upset plots[58], which were generated with ComplexUpset[59].

For the analyses displayed in Fig. 3 processing results were integrated across the different LC-MS setups merging the processing results (from the analyses described above for each dilution level and species). Intensities for each protein were aggregated by calculating the mean and normalized against the maximum reported protein intensity value within each LC-MS setup. These normalized values were then combined across labs for each PYE dilution to obtain a single intensity value per protein, which was then ranked (Fig. 3a, b). For the scatter plot analysis, protein intensities were averaged and normalized separately for each LC-MS setup and PYE dilution level, to assess and plot the correlation between protein intensities across the different PYE dilution levels, i.e., PYE sample sets (Fig. 3c, d). To this end, we divided the LFQ values for each protein by the LFQ value of the most abundant protein (highest LFQ value) for each site and setup. Ratio were then multiplied by 100 to convert into percent, with 100% corresponding to the highest LFQ value. Figure subpanels have been integrated using Adobe Illustrator (version 29.7.1). Bar plots in Figs. 1 and 7 have been generated using GraphPad Prism (version 10.5.0).

## Ethics statement

Blood samples were taken at the University Medical Center of the Johannes Gutenberg University Mainz from five healthy donors after obtaining informed consent. All experiments containing human blood plasma from these donors were approved by the ethics committee of the Landesärztekammer Rheinland-Pfalz, Mainz No. 837.439.12 (8540-F) and thus performed in compliance with all relevant laws and guidelines.

## Reporting summary

Further information on research design is available in the Nature Portfolio Reporting Summary linked to this article.

## Data availability

The raw mass spectrometry data generated in this study along with the database search results have been deposited to the ProteomeXchange Consortium (http://proteomecentral.proteomexchange.org) via the jPOST partner repository[60] with the dataset identifiers PXD056598 (ProteomeXchange) [https://proteomecentral.proteomexchange.org/cgi/GetDataset?ID=PXD056598] and JPST003358 (jPOST, https://repository.jpostdb.org/entry/JPST003358) (PYE analyses from all partner sites as well as plasma proteome experiments). An overview of deposited data files is also provided in Supplementary Data 9. Source data are provided with this paper via Zenodo at [https://doi.org/10.5281/zenodo.17131745]. Additional data files providing a full summary of identified proteins and peptides across all sites for the DDA and DIA analyses can be also assessed via Zenodo at [https://doi.org/10.5281/zenodo.17131745]. Source data are provided with this paper.

## Code availability

The R scripts for reproducing the figures are available via GitHub at [https://github.com/HanYoo1402/LFQ-Bench-Scripts-for-PYE-Multicenter-Study and Zenodo at [https://doi.org/10.5281/zenodo.17018339].

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

## Acknowledgements

We thank Christina Jung for excellent technical assistance at the DiaSym study center and Elena Kumm for her assistance in sample preparation. This work was supported by the German Ministry of Education and Research (BMBF) as part of the National Research Initiative Mass Spectrometry in Systems Medicine (MSCoreSys), under the following grant agreement numbers: CLINSPECT-M [FKZ 03LW0248 and FKZ 161L0214E to S.M.H., FKZ 161L0214A and 16LW0243K to B.K., J.T., FKZ 161L0214C to A.I., B.K., S.F.L.], SMART-CARE [FKZ 161L0213 to J.K., SMART-CARE 031L0212B, SMART-CARE2 16LW0234 to U.K.], MSTARS [01EP2201 to M.R. and 16LW0239K to M.M.], CurATime [diAMs, FKZ 03ZU1202EA to S.T.] and DIASyM [FKZ 031L0241A/B to S.T.], DIASyM2 [FKZ 03LW0241K to S.T.] as well as the BMBF LiSyM-Cancer networks SMART-NAFLD 031L0256A and C-TIP-HCC 031L0257C and the German Center for Lung Research, DZL3.0 82DZL004B4 and DZL4.0 82DZL004C4 to U.K. Additionally, we acknowledge FOR 5146, by HORIZON EUROPE of the European Research Council within the network ARTEMIS 101136299 funded to U.K. This work was further funded by the German Research Foundation as follows: DFG SFB1066 (TP-Q6 to S.T.), SFB1292/2 (project number 318346496, TP11 to U.D., and TP-Q01 to S.T.); the DFG priority program SPP 2225 (grant number 446605368 to U.D.) and the DFG Germany's Excellence Strategy within the framework of the Munich Cluster for Systems Neurology (EXC 2145 SyNergy – project number 390857198 to S.F.L.). The BayBioMS, BayBioMS@MRI and Charité core facility mass spectrometers were funded in part by the German Research Foundation: INST 95/1435-1 FUGG (Exploris 480) and INST 95/1436-1 FUGG (Orbitrap Fusion Lumos) to BayBioMS; INST 95/1649-1 FUGG (Exploris 480) and INST 95/1650-1 FUGG (Orbitrap Eclipse) to BayBioMS@MRI; grant number 492697668 (zenoTOF) to the Core Facility of Mass Spectrometry at the Charité. This work was further supported by the Research Center for Immunotherapy (FZI) of the Johannes Gutenberg-University Mainz.

## Author contributions

U.D. and S.T. conceived and supervised the study. M. Scherer, C. Leps, D. Hein, U.D., prepared and distributed samples, U.D., O.K., M. Sielaff, A.M.J., D.G.Z., C.T., J.M.P., T.K.B., J.T., P.G., T.M., G.K., K.A., B.H., H.U., D.L.F., D. Helm, L.S., O.P., D.Q., S.I.W., L.R.S., J.M., C. Ludwig. conducted mass spectrometric analyses. U.D., H.B.Y., M. Sielaff, D. Hein, analysed the data, U.D. and H.B.Y. generated figures and prepared the initial draft of the manuscript. O.K., M. Sielaff, A.M.J., D.G.Z., T.K.B., J.T., P.G., K.A., B.H., H.U., D.L.F., D. Helm, L.R.S., J.M., C. Ludwig, A.I., B.K., S.F.L., J.K., U.K., P.M., F.C., M.R., M.M., S.M.H., S.T. discussed results and contributed to writing. All authors reviewed the final manuscript version.

## Funding

## Competing interests
T.M. and G.K. are employees of Bruker. B.K. is a co-founder and shareholder of OmicScouts and MSAID. He has no operational role in either company. The remaining authors declare no competing interests.

## Additional information

Ute Distler [1,2] ✉, Han Byul Yoo[1,2], Oliver Kardell[3], Dana Hein[1,2], Malte Sielaff[1,2], Marian Scherer[1,2], Anna M. Jozefowicz[1,2], Christian Leps[1,2], David Gomez-Zepeda [4,5], Christine von Toerne [3], Juliane Merl-Pham [3], Teresa K. Barth[6], Johanna Tüshaus [7], Pieter Giesbertz [8,9], Torsten Müller[10,11], Georg Kliewer[10,11], Karim Aljakouch[10,11], Barbara Helm[12,13], Henry Unger [12,14], Dario L. Frey [12,13,15], Dominic Helm [14,15], Luisa Schwarzmüller [15], Oliver Popp[16], Di Qin[17], Susanne I. Wudy [18], Ludwig Roman Sinn [19,20], Julia Mergner [21], Christina Ludwig [18], Axel Imhof [6], Bernhard Kuster [7,18], Stefan F. Lichtenthaler [8,9,22], Jeroen Krijgsveld[10,11], Ursula Klingmüller[12,13,14,23], Philipp Mertins [16], Fabian Coscia [17], Markus Ralser [19], Michael Mülleder [20], Stefanie M. Hauck [3] & Stefan Tenzer [1,2,4,5] ✉

[1]Institute of Immunology, University Medical Center of the Johannes Gutenberg University Mainz, Mainz, Germany. [2]Research Center for Immunotherapy (FZI), University Medical Center of the Johannes Gutenberg University Mainz, Mainz, Germany. [3]Metabolomics and Proteomics Core, Helmholtz Zentrum München, German Research Center for Environmental Health, Munich, Germany. [4]German Cancer Research Center (DKFZ), Heidelberg, Germany. [5]Immunoproteomics Unit, Helmholtz-Institute for Translational Oncology (HI-TRON) Mainz, Mainz, Germany. [6]Clinical Protein Analysis Unit (ClinZfP), Biomedical Center, Faculty of Medicine, LMU Munich, Munich, Germany. [7]Chair of Proteomics and Bioanalytics, Technical University of Munich, Freising, Germany. [8]German Center for Neurodegenerative Diseases (DZNE) Munich, DZNE, Munich, Germany. [9]Neuroproteomics, School of Medicine and Health, Klinikum rechts der Isar, Technical University of Munich, Munich, Germany. [10]Division of Proteomics of Stem Cells and Cancer, German Cancer Research Center (DKFZ), Heidelberg, Germany. [11]Medical Faculty, Heidelberg University, Heidelberg, Germany. [12]Division Systems Biology of Signal Transduction, German Cancer Research Center (DKFZ), Member of the German Center for Lung Research (DZL), Heidelberg, Germany. [13]German Center for Lung Research (DZL) and Translational Lung Research Center Heidelberg (TLRC), Heidelberg, Germany. [14]Liver Systems Medicine against Cancer (LiSyM-Krebs), Heidelberg, Germany. [15]Proteomics Core Facility, German Cancer Research Center (DKFZ), Heidelberg, Germany. [16]Max-Delbrück-Center for Molecular Medicine in the Helmholtz Association (MDC), Berlin, Germany. [17]Max-Delbrück-Center for Molecular Medicine in the Helmholtz Association (MDC), Spatial Proteomics Group, Berlin, Germany. [18]Bavarian Center for Biomolecular Mass Spectrometry (BayBioMS), TUM School of Life Sciences, Technical University of Munich, Freising, Germany. [19]Department of Biochemistry, Charité Universitätsmedizin Berlin, Berlin, Germany. [20]Core Facility High-Throughput Mass Spectrometry, Charité Universitätsmedizin, Berlin, Germany. [21]Bavarian Center for Biomolecular Mass Spectrometry at Klinikum rechts der Isar (BayBioMS@MRI), TUM School of Medicine and Health, Technical University of Munich, Munich, Germany. [22]Munich Cluster for Systems Neurology (SyNergy), Munich, Germany. [23]German Consortium for Translational Cancer Research (DKTK), Heidelberg, Germany. ✉e-mail: ute.distler@uni-mainz.de; tenzer@uni-mainz.de

