## [Transparent Peer Review file · Nature Communications]

Multicenter Evaluation of Label-Free Quantification in Human Plasma on a High Dynamic Range Benchmark Set

Corresponding Author: Dr Ute Distler

Version 0:

Reviewer comments:

Reviewer #1

(Remarks to the Author)

The authors present a multi-center ring-trial study on serum proteomics with additional spike-ins to assess relative protein quantitation. Particular highlights are the large number of sites and the fact that each site used their preferred measurement protocol. Yet, the study shows very good concordance of the results, highlighting robustness of mass-spectrometry-based proteomics. In addition, the comparison of DIA vs DDA based data acquisition is helpful.

I only have one major comment: in my view, the study would benefit from including serum samples lacking the yeast/bacterial spike-ins. This would allow to assess how "true" missingness is addressed, e.g. by the data analysis software. It is probably not feasible to include this aspect for all partner sites but it would be helpful to see this for two or three sites.

I would also appreciate an in-depth Supplementary Table that directly compares the human peptide and protein IDs across all sites, with all sequences and scores. I hope not to have overlooked such a table.

Reviewer #2

(Remarks to the Author)

In this manuscript, the authors present the results of a multi-center study for evaluation of mass spectrometry methods to analyze proteins in unfractionated plasma. A variety of different instrument configurations were used (called "setups" in the manuscript). Overall, the results are very interesting, and document that data-independent acquisition (DIA) MS provided a larger number of protein identifications and more accurate relative quantification than data-dependent acquisition (DDA) MS. This finding will not be a surprise to mass spectrometrists who use both techniques, but, nevertheless, the information that is presented will be valuable through insight into factors that contribute to enhanced results. For those who have not used DIA-MS, the manuscript should help to encourage them to try it. However substantial revision of the manuscript is needed before publication can be recommended. Specific comments about items that need clarification or alteration are included as "sticky notes" in an annotated version of the manuscript. More general points are discussed below.

The first problem I had with the manuscript was the description of the samples that were used for the analyses. It was clear in the beginning of the Results section that low levels of yeast (Y) and E. coli (E) tryptic peptides had been added to digests of human plasma proteins (P), and that stepwise lower concentrations of Y and E were achieved through addition of more plasma digest. However, the design used to obtain various proportions of Y and E was somewhat convoluted, and the information in section 2.1 provides only a general description. In particular, it was not clear until the Methods section that samples "A" and "B" for each dilution were not replicates. To me the sample arrangement was unduly complicated and was not totally necessary to achieve the study goals. While the samples, obviously, can't be changed at this stage, the descriptions of the relative proportions of the components need to be clear as early as possible. (As indicated in a "sticky note" in Figure 1, the sample compositions should be directly understandable in the overview. And, the panels that are intended to represent the results should not be included in the figure since they do not provide useful information.)

The second problem relates to the figures in that the majority do not clearly convey the point(s) being made. An example is Figure 3 entitled "Protein dynamic range and protein intensity distribution across the full PYE sample set integrating data

from all sites." In addition to the fact that the colors in each panel are difficult to distinguish, the trends being shown are not obvious. One immediate question is why are the values on the y-axis negative since they are supposed to be derived from intensities? What do the plots in Figure 3 panels A and B indicate? In Figure 4, the box plots in panels A and B are clear representations of the results, while findings being shown in the rest of the panels are not directly deducible. Since the goals of the study were clear—to find the method(s) that provided the largest number of protein IDs with the most accurate relative quantification—why not find ways to display the results in formats that readers can immediately understand? Dot, box and violin plots along with bar charts and some line graphs should be all that's needed. If desired, some of the non-standard plots could be provided as supplemental data, but not as key figures in the manuscript.

In summary, this is an interesting study with results that are likely to be of interest to a wide range of mass spectrometry-based proteomics researchers. And, the findings for plasma seem applicable to numerous other sample types. However, substantial revision of the manuscript is needed—most notably the figures—before publication can be recommended.

Reviewer #3

(Remarks to the Author)

This study assessed the performance of label-free quantitative methods (DDA and DIA) in human plasma through a multicenter approach and developed a high dynamic range multispecies sample set (PYE) to simulate the complexity of clinical plasma. Over 1,000 datasets were collected and analyzed. While this work is significant, the analysis presented is relatively superficial and lacks deeper exploration of the datasets, which limits the practical implications of the conclusions drawn. Specific comments are as follows:

1. To conduct a general experiment to verify the qualitative and quantitative accuracy of different mixed proportions of species, it is reasonable to spike Yeast and E. coli samples in the plasma. However, for developing standard samples intended for broad use, mixing Yeast and E. coli into plasma has limited significance. First, these two species have similar complexity levels, and including just one species may be sufficient to reflect the analytical challenge. Second, we believe incorporating multiple standard proteins at varying concentrations could provide more meaningful insights.
2. In this manuscript, LC-MS platforms were treated as "black boxes", which limits the depth and persuasiveness of the conclusions drawn. For the PYE samples, the authors utilized different acquisition strategies (DDA versus DIA), different MS instruments, and varied parameter settings. This approach has led to discrepancies in the number of identifications. Specifically, these variations influence the final identification results by affecting factors such as the number of acquired MS2 spectra, spectra quality, and peptide repeated acquisition rate. It is important to evaluate whether LC-MS parameter settings from certain platforms may have inherent limitations that contribute to suboptimal results. Why are there differences in the number of identification results on LC-MS platforms of the same type? Furthermore, it would be beneficial for the authors to provide a data acquisition guideline tailored to current mass spectrometry technology, particularly for plasma samples.
3. A number of studies have demonstrated that DIA outperforms DDA. While it is straightforward to compare qualitative and quantitative performance metrics, a more detailed explanation is warranted regarding the specific reasons why DIA data for plasma samples surpasses DDA data.
4. In this study, the data were analyzed using only two software tools: MaxQuant for DDA data and DIANN for DIA data. To enhance the robustness of the analysis, we propose incorporating results from additional software tools. Specifically, when comparing peptide and protein counts, it is crucial to provide a detailed description to ensure the comparability of results across different software. For instance, for a specific set of peptides, the number of protein groups identified can vary significantly depending on the protein inference algorithm used. Therefore, it is important to note that a higher number of identified protein groups does not necessarily indicate superior performance, while the reported peptides are the same.
5. While we concur with the fundamental conclusions presented in this study, our experience suggests that the missing values in the plasma DDA data appears unusually high. We therefore request that the original MaxQuant results and detailed descriptions of the subsequent analysis be provided in the supplementary materials.
6. We recommend that the authors summarize previously published representative studies in plasma proteomics to contextualize and highlight the technical advancements achieved in this study.
7. It is appreciated that the authors replenish analysis results obtained from the latest LC-MS platforms, such as the Astral, to further validate and enhance the robustness of their findings.

Version 1:

Reviewer comments:

Reviewer #1

(Remarks to the Author)

All comments have been adequately addressed. I congratulate the authors on their nice work.

Reviewer #2

(Remarks to the Author)

The authors have provided comprehensive responses to my comments/questions about the original version of this manuscript and have revised their manuscript accordingly. While there are a few relatively minor items that would still benefit from revision/clarification, they are not critical enough to warrant another round of revision and review. The manuscript is very "dense" with extensive details about the large number of analyses that were conducted. It is likely to be a valuable resource with benchmarks for human plasma proteomics. [Note: it would have been very helpful if the figure numbers would have been shown on the corresponding pages (or were they there and I missed them?)]

Reviewer #3

(Remarks to the Author)

The authors are sincerely appreciated for their agreement of the raised issues and their professional responses. This study presents a systematic evaluation and comprehensive analysis of plasma proteomics. Plasma proteomics analysis still has great potential for development, particularly when compared with recent advancements in single-cell proteomics over the past one to two years. There is considerable anticipation for the identification of plasma proteomics to reach the level of thousands of proteins. In conclusion, this study has pushed the boundaries of existing analytical and evaluative techniques to their limits, and its publication is expected to provide significant reference value for researchers in the field.

Reviewer #1 (Remarks to the Author):

The authors present a multi-center ring-trial study on serum proteomics with additional spike-ins to assess relative protein quantitation. Particular highlights are the large number of sites and the fact that each site used their preferred measurement protocol. Yet, the study shows very good concordance of the results, highlighting robustness of mass-spectrometry-based proteomics. In addition, the comparison of DIA vs DDA based data acquisition is helpful.

We thank the reviewer for this positive feedback.

I only have one major comment: in my view, the study would benefit from including serum samples lacking the yeast/bacterial spike-ins. This would allow to assess how “true” missingness is addressed, e.g. by the data analysis software. It is probably not feasible to include this aspect for all partner sites but it would be helpful to see this for two or three sites.

We thank the reviewer for the comment and this suggestion. We agree that the study would benefit from a neat plasma analysis across different laboratories without spike-ins. We generated a new plasma digest, which was measured at three different sites (laboratories G, H and L) using the same set-ups as for the round robin study (Orbitrap platforms, DIA mode, six replicate injections). In total, these remeasurements comprise data from four different LC-MS setups (G_ulti_ex, L_ulti_ex, H_ulti_ex, H_ulti_ecl). Additionally, to assess interlaboratory reproducibility, we analysed the neat plasma sample in lab L using the same Ultimate/Exploris 480 setups as laboratories G and H. Across the six datasets numbers of identified proteins ranged between 276 to 428. Overall, we found a high overlap of identified proteins in neat plasma across the different setups and sites (Fig. 7F,G), with 235 proteins identified across all setups, and around 300 proteins when the setup with the lowest number of identifications is not taken into account (Fig. 7G, Reviewer Fig. 1A). We now integrate this data in the main manuscript (Fig. 7F,G).

Moreover, we further exploited these results to assess the effectiveness of the false discovery rate (FDR) control using an entrapment strategy¹ (see Reviewer Fig. 1B,C). To this end, we searched the neat plasma sample against a custom compiled database containing the (reviewed) human, yeast and *E. coli* reference proteomes and common contaminants to estimate the number of false positive identifications allowing us to assess the number of wrongly assigned proteins not present in the dataset. Summarizing the results of the unfiltered DIA-NN report, we retrieved a rather high number of false positives – between 4.7 – 11 % on protein level (see Reviewer Fig. 1B). For data processing, precursor FDR was set to 1% using the in-built option in the DIA-NN GUI. However, the round robin data had been additionally filtered as follows: Reversed sequences (i.e., decoy hits), potential contaminants and proteins identified by only one peptide were removed. Q.Value, PG.Q.Value, Lib.Q.Value, and Lib.PG.Q.Value had to be below or equal 0.01 and PG.MaxLFQ > 0. Applying respective filter criteria (see Reviewer Fig. 1C) markedly reduced the percentage of false-positive identifications, which dropped to below 0.6 % indicating a low number of false-positives or “true” missingness also for the round robin dataset.

I would also appreciate an in-depth Supplementary Table that directly compares the human peptide and protein IDs across all sites, with all sequences and scores. I hope not to have overlooked such a table.

We agree with the reviewer that such a table is beneficial for comparative analyses. Upon initial submission, the “individual” search results were provided on a repository (jPOST/ProteomeXchange). We now added in-depth summary tables (Supplementary Tables 9 and 10) integrating the information across all sites on protein and peptide level (IDs, LFQ values, scores) for MaxQuant and DIA-NN. Additionally, we provide an overview table indexing all the search results uploaded to the

jPOST/ProteomeXchange repositories facilitating the re-analysis of the datasets (Supplementary Table 11).

Reviewer Fig. 1: Assessment of missing values and false positives in neat human plasma analyzed across three different labs and four different LC-MS set-ups. (A) Missing value plot. Data derived from the LC-MS analysis of tryptic plasma were processed separately (i.e., for each laboratory and LC-MS setup, six replicate injections) in DIA-NN 1.8.1 searching against the human reference proteome. Proteins were ranked according to their detection frequency across the different datasets, with the highest-ranking proteins identified in all datasets without any missing values. **(B)** Number of identified human (black bar) and *E. coli*/yeast protein groups (light gray bar) in a tryptic human plasma sample without any spike-ins after data processing in DIA-NN v1.8.1 searching against a custom compiled database containing the (reviewed) human, yeast, *E. coli* reference proteomes and common contaminants (precursor FDR set to 1%). Bar plot summarizes results from DIA-NN on protein level without any additional filters applied. **(C)** Number of identified human (black bar) and *E. coli*/yeast protein groups (light gray bar) in a tryptic human plasma sample without any spike-ins after DIA-NN v1.8.1 processing as in (B) applying additional filter criteria: Proteins have to be identified by two or more peptides, decoy hits and potential contaminants are filtered out, Q.Value, PG.Q.Value, Lib.Q.Value, and Lib.PG.Q.Value have to be below or equal 0.01 and PG.MaxLFQ > 0.

Reviewer #2 (Remarks to the Author):

In this manuscript, the authors present the results of a multi-center study for evaluation of mass spectrometry methods to analyze proteins in unfractionated plasma. A variety of different instrument configurations were used (called "setups" in the manuscript). Overall, the results are very interesting, and document that data-independent acquisition (DIA) MS provided a larger number of protein identifications and more accurate relative quantification than data-dependent acquisition (DDA) MS. This finding will not be a surprise to mass spectrometrists who use both techniques, but, nevertheless, the information that is presented will be valuable through insight into factors that contribute to enhanced results. For those who have not used DIA-MS, the manuscript should help to encourage them to try it. However substantial revision of the manuscript is needed before publication can be recommended. Specific comments about items that need clarification or alteration are included as "sticky notes" in an annotated version of the manuscript. More general points are discussed below.

We thank the reviewer for this assessment and acknowledging the value of our study.

A point-to-point answer to the sticky notes is provided in the annotated version of the manuscript in addition to the answers provided below which address the more general points raised by the reviewer.

The first problem I had with the manuscript was the description of the samples that were used for the analyses. It was clear in the beginning of the Results section that low levels of yeast (Y) and *E. coli* (E) tryptic peptides had been added to digests of human plasma proteins (P), and that stepwise lower concentrations of Y and E were achieved through addition of more plasma digest. However, the design used to obtain various proportions of Y and E was somewhat convoluted, and the information in section 2.1 provides only a general description. In particular, it was not clear until the Methods section that samples "A" and "B" for each dilution were not replicates. To me the sample arrangement was unduly complicated and was not totally necessary to achieve the study goals. While the samples, obviously, can't be changed at this stage, the descriptions of the relative proportions of the components need to be clear as early as possible. (As indicated in a "sticky note" in Figure 1, the sample compositions should be directly understandable in the overview. And, the panels that are intended to represent the results should not be included in the figure since they do not provide useful information.)

We thank the reviewer for pointing this out and apologize that the sample description was not clear in the main text at the start of the result section. We now rephrased this part (see page 7, first paragraph) to better explain the sample composition. Also, we slightly enlarged the first panel in Fig. 1A setting a stronger focus on this part of the figure. As requested by the reviewer, we also removed the "log-plot" in Fig. 1A.

We think that the reviewer's suggestion to use a different nomenclature with decreasing numbers according to the level of dilution would work well for less complex mixed proteome sample sets. Nevertheless, we opted to follow the naming for "LFQbench-type" sample sets that has been introduced in 2016 by Navarro *et al.*² following the designation HYE A and B for the two samples being compared to each other with differing amounts of spike-ins. The letters "HYE" designate species (H: human, Y: yeast, E: *E. coli*) and the letters "A" and "B" two samples that are used for comparative analyses containing different spike-in amounts (of full proteome digests). This naming scheme can be found in other benchmark studies as well³⁻⁵, and has several advantages, as described in the following paragraph.

Other studies use a different scheme where letters indicate the species and numbers, for example, sample composition (i.e., spike-in amount in percent)⁶. However, their sample set and study design follow a slightly different rationale compared to the present study. While in these studies also more than two mixed proteome samples with different "spike-in" amounts were used, they don't derive, to

our knowledge, from a direct dilution of an initial set. Moreover, spike-ins in most studies don't fall below 1% of the total protein mass. Including this information in the name itself (e.g., PYE 98.8:0.21:0.89 or P98.8Y0.21E0.89), would, to our opinion, rather convolute the sample designation instead of making it more clear. The short sample designation (indicating the dilution level – i.e., PYE3 for a 1:3 and PYE9 for a 1:9 dilution) has proven to be very efficient and useful for intra- and interlaboratory communication in this multicenter study. As samples PYE3 and PYE9 derive from a dilution of PYE1 A and B, relative abundances between samples A and B are conserved across the dilutions. As a result, expected protein ratios between A and B are the same for PYE1, PYE3 and PYE9. Hence, we feel that the actual “% of total protein mass”-designation is not crucial in the sample name itself, as $\log_2(\text{FC})$ ratios for the comparisons conducted in the present study are conserved.

The second problem relates to the figures in that the majority do not clearly convey the point(s) being made. An example is Figure 3 entitled "Protein dynamic range and protein intensity distribution across the full PYE sample set integrating data from all sites." In addition to the fact that the colors in each panel are difficult to distinguish, the trends being shown are not obvious. One immediate question is why are the values on the y-axis negative since they are supposed to be derived from intensities? What do the plots in Figure 3 panels A and B indicate? In Figure 4, the box plots in panels A and B are clear representations of the results, while findings being shown in the rest of the panels are not directly deducible. Since the goals of the study were clear—to find the method(s) that provided the largest number of protein IDs with the most accurate relative quantification—why not find ways to display the results in formats that readers can immediately understand? Dot, box and violin plots along with bar charts and some line graphs should be all that's needed. If desired, some of the non-standard plots could be provided as supplemental data, but not as key figures in the manuscript.

We appreciate this feedback and apologize if some of the plots did not convey their message clearly. We now changed several plots throughout the manuscript facilitating interpretation.

Figure 3: Initially, we normalized all intensity values (as the same software tool reports very different LFQ abundances for different instrument platforms, e.g., Exploris vs tTOF). Hence, for each site and setup we divided the LFQ values for each protein by the LFQ value of the most abundant protein (highest LFQ value). Subsequently, we calculated the logarithm (base 10) for these ratios, which mathematically results in negative values. However, we now changed this to percent (i.e., with 100% corresponding to the highest LFQ value) to avoid a negative scale. We hope this makes the plot more clear. We also extended and adapted the description in the Methods Section accordingly (see page 24). The plots in Figure 3 panels A and B depict the protein dynamic range that can be covered by the MS-based workflows used in the present study. In our opinion, they provide valuable information as they summarize and integrate this information across all laboratories and additionally provide the information on the total numbers of identified proteins across all DIA and DDA runs in this study (for each species and PYE “dilution”). As plasma analysis is particularly challenging due to the high dynamic range of plasma proteins plots in Figure 3 additionally help to determine limits of detection (LOD) for protein identification as mentioned page 10, lines 321 ff (main manuscript). Additionally, each of the original correlation plots (Fig. 3C,D) was split into two separate plots displaying the values for yeast and *E. coli* separately. This way values of one species are not convoluted by the values of the other species. Moreover, we added a line indicating the expected ratio (correlation) of abundance values between the samples that were compared as well as the calculated R^2 . Hence, those plots are now more informative and convey their message better.

Figure 4: As suggested by the reviewer, we now simplified panels C,D by changing the line/density plots to box plots avoiding the overly complex colour scheme. Additionally, we changed panels E,F to a two-colour scheme (referring to nano- and microflow setups) as the site information is not

necessarily required for the analysis conducted. Information on the performance of each individual setup is provided now in a new Supplementary Table 4. Simplifying panels C-F removed also redundant figure legends.

Figure 6: We moved the log-plots to the Supplementary Section. The “new” figure 6 contains the plots that were displayed in figure 7 upon initial submission. Particularly proteins in the low abundant range are more prone to less accurate and/or precise quantification, which can also be nicely deduced from the log-plots. Hence, as suggested by the reviewer, we opted to display additional violin plots instead in the revised version of the manuscript that integrate the LFQ values from low abundant proteins, i.e., the tertile of the dataset encompassing the proteins with the lowest abundance values (as new Fig. panels 6C,D). Corresponding violin plots for PYE3 are provided in Supplementary Fig. 12.

In general, we now added more references in the text, particularly in the figure legends, pointing to the Supplementary Tables that provide either additional information or source data for the different analyses displayed.

In summary, this is an interesting study with results that are likely to be of interest to a wide range of mass spectrometry-based proteomics researchers. And, the findings for plasma seem applicable to numerous other sample types. However, substantial revision of the manuscript is needed—most notably the figures—before publication can be recommended.

We thank the reviewer for this positive feedback and all the suggestions which helped us to improve the manuscript. We hope the reviewer finds the revised version to be clearer and more comprehensible.

Reviewer #3 (Remarks to the Author):

This study assessed the performance of label-free quantitative methods (DDA and DIA) in human plasma through a multicenter approach and developed a high dynamic range multispecies sample set (PYE) to simulate the complexity of clinical plasma. Over 1,000 datasets were collected and analyzed. While this work is significant, the analysis presented is relatively superficial and lacks deeper exploration of the datasets, which limits the practical implications of the conclusions drawn. Specific comments are as follows:

We thank the reviewer the positive comment about the significance of our work. We have deepened the exploration of the datasets, and respond to the specific comments below.

To conduct a general experiment to verify the qualitative and quantitative accuracy of different mixed proportions of species, it is reasonable to spike Yeast and *E. coli* samples in the plasma. However, for developing standard samples intended for broad use, mixing Yeast and *E. coli* into plasma has limited significance. First, these two species have similar complexity levels, and including just one species may be sufficient to reflect the analytical challenge.

We agree that the PYE samples have certain limitations, but the broad acceptance and adoption of mixed proteome samples by the community has proven the significance of this strategy. Indeed, since the introduction of mixed proteome samples for the benchmarking of instrument platforms and processing software, this concept has been widely used by many laboratories and instrument manufacturers.

As intended by design, *E. coli* and yeast have similar complexity levels, both providing thousands of peptides present at exactly defined ratios between samples. This large number of analytes with defined ratios is almost impossible to obtain with approaches spiking in only a limited number of proteins. As a result, the datasets from "LFQbench-type" experimental designs provide a more robust estimate of the distribution of errors which better resembles natural samples.

Here, we decided to use a sample set with three distinct proteomes since it provides several advantages over simpler two-proteome samples. First, this sample set enables the evaluation of both „upregulated“ and “downregulated“ analytes (i.e., tryptic peptides) with known abundance ratios. Second, this also helps to avoid issues with normalization. Additionally, spiking two proteomes into plasma also increases the complexity of the sample across the entire dynamic range, thus challenging the analytical capabilities of the tested platforms.

Second, we believe incorporating multiple standard proteins at varying concentrations could provide more meaningful insights.

We agree that spiking of defined standard proteins has certain advantages, including the option to define a wide range of spike-in levels, which is particularly valuable to determine the linear range of quantification. However, this approach also has certain shortcomings, including the relatively low number of analytes being assessed.

One advantage of the design of the PYE sample is that it allows to compare LFQ values of *E. coli* spike-ins across six dilution levels and multiple abundance ranges. As shown in our new Figure 3E,F, the extended “LFQbench” design used in this manuscript enables to precisely determine limits of detection (LOD) and linearity for thousands of analytes as a function of their relative signal intensities. In contrast to a limited number of “spiked analytes”, this allows to comprehensively analyse the statistics of missing values as a function of signal intensity.

2. In this manuscript, LC-MS platforms were treated as "black boxes", which limits the depth and persuasiveness of the conclusions drawn. For the PYE samples, the authors utilized different

acquisition strategies (DDA versus DIA), different MS instruments, and varied parameter settings. This approach has led to discrepancies in the number of identifications. Specifically, these variations influence the final identification results by affecting factors such as the number of acquired MS2 spectra, spectra quality, and peptide repeated acquisition rate. It is important to evaluate whether LC-MS parameter settings from certain platforms may have inherent limitations that contribute to suboptimal results. Why are there differences in the number of identification results on LC-MS platforms of the same type? Furthermore, it would be beneficial for the authors to provide a data acquisition guideline tailored to current mass spectrometry technology, particularly for plasma samples.

We fully agree with the reviewer that different acquisition strategies, instruments and varied parameters markedly impact the number of identifications posing a challenge in the field of clinical proteomics. While many mass spectrometry laboratories worldwide focus on clinical proteomics applications and biomarker discovery, they have different equipment, financial leeway, SOPs and workflows. Hence, the primary goal of the present multicenter study was to assess the degree of similarities as well as the differences in a more “real-life” scenario where each laboratory uses their own preferred workflow with the equipment available as opposed to selecting one workflow or LC-MS setup over the other. We could show, that irrespective of the setup used, a high overlap in identifications and, particularly for the DIA approaches, good quantitative performance can be achieved. Of note, for all DDA analyses (except the one from laboratory F) there is a corresponding DIA analysis that was conducted in the same laboratory, on the same LC-MS platform, with the exact same LC settings (gradient, column, etc) only differing in the acquisition mode (see also Table 1 and Supplementary Table 1). Hence, this allows to directly assess the influence of the acquisition mode for these setups as discussed page 12, second paragraph (Figure 5). We rephrased this part (lines 373-375) to highlight that a back-to-back comparison of DDA and DIA is possible for these setups as the exact same LC-MS setup at the same site was used.

We also agree with the reviewer that the manuscript would benefit from a more in-depth investigation on how the number of identifications are influenced on a distinct platform for the DIA analyses facilitating also the decision process for adapting distinct parameters (see new result section 2.5, page 15): The “ulti_ex” and “nLC_tTOF” DIA setups from laboratories H and G provided the highest proteome coverage. We remeasured the PYE1 sample replicating the respective LC-MS settings in laboratory L identifying very similar numbers of proteins and peptides (Fig. 7A) with a high overlap (Fig. 7B) demonstrating the interlaboratory transferability of methods (even two years after the initial measurements). Moreover, we could also demonstrate that the quantitative pattern follows the same distributions as compared to the initial round robin data (Fig. 7C). Those measurements also include the “ulti_ex” setup from laboratory G, as this is the exact same setup as for H (same LC-MS platform and column setup but half the analysis time – 60 min vs 120 min and slightly different DIA method with adapted lower cycle time, see Supplementary Table 1). In line with the round robin data, remeasurements in laboratory L also provided lower proteome coverage for the “G_ulti_ex” setup.

To better understand the influence of LC-MS parameters we additionally compared the influence of the MS method and chromatographic setup for the EASY-nLC 1200 – timsTOF setup of laboratory G (“G_nLC_TOF”), which used an IonOpticks Aurora column (75 μ m ID x 25 cm) for peptide separation running a 30 min gradient at 300 nL/min. To this end, we analyzed the PYE1 sample using the MS method of laboratory G but the LC setting from laboratory L (Bruker PepSep setup, 150 μ m ID x 25 cm, 35.5 min gradient at 850 nL/min). This resulted in a marked drop in the number of identified proteins and peptides (see Fig. 7D, here a more detailed description of LC-MS parameters is provided as well), while the MS method (30 Da vs 25 Da fixed window scheme, different IMS range and cycle time) did not affect the number of identifications. These data clearly demonstrate, that the LC setup used by laboratory G in the round robin study (IonOpticks Aurora column, 75 μ m ID x 25 cm, with a 30 min gradient at 300 nL/min) outperforms the LC settings used by laboratory L (Bruker PepSep, 150 μ m ID x 25 cm, 35.5 min gradient at 850 nL/min).

In addition, we analyzed neat plasma without any spike-ins across three different sites using four different Orbitrap LC-MS setups from the round robin study (see also reply reviewer #1). Here, we also obtained similar results as compared to the measurement of the PYE samples (Fig. 7F,G): In line with the round robin study, the “H_ulti_ex” setup provided the best proteome coverage also for a neat plasma sample indicating that plasma analysis, at least on the Orbitrap platforms used in the present study benefit from a long gradient and analysis times.

3. A number of studies have demonstrated that DIA outperforms DDA. While it is straightforward to compare qualitative and quantitative performance metrics, a more detailed explanation is warranted regarding the specific reasons why DIA data for plasma samples surpasses DDA data.

In label-free quantitative proteomics, all quantitative information is obtained on precursor level for DDA type acquisitions. DIA approaches provide a more comprehensive sampling of the fragment ion space, thus enhancing the detectability of lower abundance fragment ions and also enable quantification at MS2 level.

Another factor is the choice of processing software, we used MaxQuant and DIA-NN in our initial study, and have now added additional data processing results from FragPipe for both DDA and DIA. Those data indicate a smaller gap in proteome coverage, while they confirm the advantage of DIA in terms of reproducibility and precision.

We now also include this in the discussion in the main text, pages 9 and DIA discussion (FragPipe results and software impact). We thank the reviewer for highlighting this added value of our study.

4. In this study, the data were analyzed using only two software tools: MaxQuant for DDA data and DIANN for DIA data. To enhance the robustness of the analysis, we propose incorporating results from additional software tools. Specifically, when comparing peptide and protein counts, it is crucial to provide a detailed description to ensure the comparability of results across different software. For instance, for a specific set of peptides, the number of protein groups identified can vary significantly depending on the protein inference algorithm used. Therefore, it is important to note that a higher number of identified protein groups does not necessarily indicate superior performance, while the reported peptides are the same.

We fully agree with the reviewer that the software tool and processing settings markedly impact the results and reported protein groups depend on the protein inference algorithm used. In the initial submission, we provided also numbers and overlap of identified peptides (Supplementary Figure 1 and 2). However, we agree with the reviewer that a more comprehensive list of identified peptides and proteins compiling results across all laboratories should be provided (see also comment reviewer #1 and point below).

Initially, we opted for MaxQuant and DIA-NN to follow-up on a previous multicenter longitudinal study on plasma/serum proteomics from the MScoreSys consortium⁷ published earlier this year. Also, the major scope of the study was not to compare the performance of different software tools for plasma analysis but rather evaluate interlaboratory and platform comparability. Nevertheless, we fully agree with the reviewer that it is a more fair comparison between DDA and DIA data if both datasets were analysed with the same tool. Hence, we now additionally processed the whole PYE dataset with the latest version of FragPipe (v23). Of note, we first processed the full dataset with FragPipe v22 as v23 was not released yet, i.e., when we started compiling the data for revision. However, using normalization resulted in some strange effects for the DDA dataset (see Reviewer Fig. 2).

Upon consultation with the developers, we reprocessed the dataset using FragPipe v23 without cross-run normalization, which was disabled using the --no-norm command (see also Material and Methods section). For better comparability with the DDA data, cross-run normalization was also switched off for DIA processing. Testing processing settings with and without cross-run normalization for DIA led

to similar results regarding global precision and accuracy. However, CVs of protein abundances between replicate injections markedly improve using normalization (see Reviewer Fig. 3). We include and discuss the FragPipe v23 results (non-normalized data) in the revised version of the manuscript (see Supplementary Figs. 4-7 and main text page 9).

Reviewer Fig. 2: Reanalysis of the PYE1 dataset from the multicenter study using FragPipe. Data from each LC-MS setup were processed separately in FragPipe version 22 using cross-run normalization. Panel (A) depicts the processing results for the DDA runs and panel (B) for the DIA analyses. Upper panels display the log-transformed ratios ($\log_2(A/B)$) of proteins plotted over the log-transformed intensity of sample A for PYE1 analyzed with the setup "A_evo_ex". Lower panels show the violin plots of log-transformed ratios ($\log_2(A/B)$) of protein abundances for the PYE1 set acquired in (A) DDA and (B) DIA mode with 14 (DDA) and 20 (DIA) different LC-MS setups, respectively. Solid lines within the violin plot indicate the median $\log_2(A/B)$ value for each setup and dashed lines the expected $\log_2(A/B)$ values for human (orange), yeast (violet), and *E. coli* (green) proteins. Data clearly demonstrate issues with the normalization in the DDA set for low abundance proteins resulting in a poor precision and accuracy quantifying *E. coli* and yeast proteins.

Reviewer Fig. 3: Comparison of coefficients of variation (CVs) of protein abundances for replicate analyses of sample PYE1 A. CVs of protein abundances processing LC-MS data with **(A)** FragPipe v22 using cross-run normalization and **(B)** FragPipe v23 without cross-run normalization. Lower panels display the log-transformed ratios ($\log_2(A/B)$) of proteins plotted over the log-transformed intensity of sample A for PYE1 analyzed with the setup “A_evo_ex”.

5. While we concur with the fundamental conclusions presented in this study, our experience suggests that the missing values in the plasma DDA data appears unusually high. We therefore request that the original MaxQuant results and detailed descriptions of the subsequent analysis be provided in the supplementary materials.

All original MaxQuant as well as DIA-NN results had been uploaded to a public repository (jPOST/ProteomXchange) upon submission. We now added an additional overview table (Supplementary Table 11) highlighting all the search results uploaded to the jPOST/ProteomXchange repositories, which now also include the FragPipe search results and new analyses (Fig. 7) facilitating re-analysis and exploration of the dataset.

In addition, we added an in-depth Supplementary Tables that directly compare peptide and protein IDs across all sites, with all sequences and scores (see also comment of reviewer #1).

As requested by the reviewer, we added more detailed descriptions of the data analyses conducted, see also Supplementary Table 8 (MaxQuant and DIA-NN parameters) and Material and Methods section “Downstream analysis of PYE data sets”.

Of note, processing the DDA data with the latest version of FragPipe resulted in a markedly higher number of identified peptides and proteins showing a much better data completeness and overlap between replicate injections and the different LC-MS setups (Supplementary Figs. 4-6).

6. We recommend that the authors summarize previously published representative studies in plasma proteomics to contextualize and highlight the technical advancements achieved in this study.

We thank the reviewer for this recommendation and apologize if we have overlooked some plasma related benchmark studies. We now include an additional paragraph in the introductory section highlighting recent developments in plasma proteomics (page 5) and further extended the discussion contextualizing the technical advancements achieved in this study.

7. It is appreciated that the authors replenish analysis results obtained from the latest LC-MS platforms, such as the Astral, to further validate and enhance the robustness of their findings.

We thank the reviewer for this suggestion. However, the data presented in this manuscript reflect the status in the different labs at the timepoint of the start of our cross-lab study. While the Astral platform has shown significant improvements in terms of proteome coverage, it was not yet available in the participating laboratories at that time.

Furthermore, a fair and unbiased benchmarking of latest generation instruments would mandate to include also other recent instrument platforms, such as the timsUltra AIP, or the ZenoTOF 8600 from Sciex, which are not available in the participating laboratories at the moment. Therefore, these novel instrument platforms could be evaluated in a follow-up manuscript.

REFERENCES

- (1) Wen, B.; Freestone, J.; Riffle, M.; MacCoss, M. J.; Noble, W. S.; Keich, U. Assessment of False Discovery Rate Control in Tandem Mass Spectrometry Analysis Using Entrapment. *Nat Methods* **2025**, 1–10. <https://doi.org/10.1038/s41592-025-02719-x>.
- (2) Navarro, P.; Kuharev, J.; Gillet, L. C.; Bernhardt, O. M.; MacLean, B.; Röst, H. L.; Tate, S. A.; Tsou, C.-C.; Reiter, L.; Distler, U.; Rosenberger, G.; Perez-Riverol, Y.; Nesvizhskii, A. I.; Aebersold, R.; Tenzer, S. A Multicenter Study Benchmarks Software Tools for Label-Free Proteome Quantification. *Nat Biotechnol* **2016**, *34* (11), 1130–1136. <https://doi.org/10.1038/nbt.3685>.
- (3) Zhang, F.; Ge, W.; Huang, L.; Li, D.; Liu, L.; Dong, Z.; Xu, L.; Ding, X.; Zhang, C.; Sun, Y.; A, J.; Gao, J.; Guo, T. A Comparative Analysis of Data Analysis Tools for Data-Independent Acquisition Mass Spectrometry. *Mol Cell Proteomics* **2023**, *22* (9), 100623. <https://doi.org/10.1016/j.mcpro.2023.100623>.
- (4) Van Puyvelde, B.; Daled, S.; Willems, S.; Gabriels, R.; Gonzalez de Peredo, A.; Chaoui, K.; Mouton-Barbosa, E.; Bouyssié, D.; Boonen, K.; Hughes, C. J.; Gethings, L. A.; Perez-Riverol, Y.; Bloomfield, N.; Tate, S.; Schiltz, O.; Martens, L.; Deforce, D.; Dhaenens, M. A Comprehensive LFQ Benchmark Dataset on Modern Day Acquisition Strategies in Proteomics. *Sci Data* **2022**, *9* (1), 126. <https://doi.org/10.1038/s41597-022-01216-6>.
- (5) Jumel, T.; Shevchenko, A. Multispecies Benchmark Analysis for LC-MS/MS Validation and Performance Evaluation in Bottom-Up Proteomics. *J Proteome Res* **2024**, *23* (2), 684–691. <https://doi.org/10.1021/acs.jproteome.3c00531>.
- (6) Guzman, U. H.; Martinez-Val, A.; Ye, Z.; Damoc, E.; Arrey, T. N.; Pashkova, A.; Renuse, S.; Denisov, E.; Petzoldt, J.; Peterson, A. C.; Harking, F.; Østergaard, O.; Rydbirk, R.; Aznar, S.; Stewart, H.; Xuan, Y.; Hermanson, D.; Horning, S.; Hock, C.; Makarov, A.; Zabrouskov, V.; Olsen, J. V. Ultra-Fast Label-Free Quantification and Comprehensive Proteome Coverage with Narrow-Window Data-Independent Acquisition. *Nat Biotechnol* **2024**, *42* (12), 1855–1866. <https://doi.org/10.1038/s41587-023-02099-7>.
- (7) Kardell, O.; Gronauer, T.; von Toerne, C.; Merl-Pham, J.; König, A.-C.; Barth, T. K.; Mergner, J.; Ludwig, C.; Tüshaus, J.; Giesbertz, P.; Breimann, S.; Schweizer, L.; Müller, T.; Kliewer, G.; Distler, U.; Gomez-Zepeda, D.; Popp, O.; Qin, D.; Teupser, D.; Cox, J.; Imhof, A.; Küster, B.; Lichtenthaler, S. F.; Krijgsveld, J.; Tenzer, S.; Mertins, P.; Coscia, F.; Hauck, S. M. Multicenter Longitudinal Quality Assessment of MS-Based Proteomics in Plasma and Serum. *J Proteome Res* **2025**, *24* (3), 1017–1029. <https://doi.org/10.1021/acs.jproteome.4c00644>.